# Enhancing the Therapeutic Effect and Bioavailability of Irradiated Silver Nanoparticle-Capped Chitosan-Coated Rosuvastatin Calcium Nanovesicles for the Treatment of Liver Cancer

**DOI:** 10.3390/pharmaceutics17010072

**Published:** 2025-01-07

**Authors:** Tamer Mohamed Mahmoud, Mohamed Mahmoud Abdelfatah, Mahmoud Mohamed Omar, Omiya Ali Hasan, Saad M. Wali, Mohamed S. El-Mofty, Mohamed G. Ewees, Amel E. Salem, Tarek I. Abd-El-Galil, Dina Mohamed Mahmoud

**Affiliations:** 1Pharmaceutics and Industrial Pharmacy Department, Al-Manara College for Medical Sciences, Maysan 62010, Iraq; tamer.nafady@pharm.mti.edu.eg; 2Department of Pharmaceutics, Faculty of Pharmacy, Nahda University, Beni Suef 62764, Egypt; dina.abdelfatah@nub.edu.eg; 3Department of Pharmaceutics and Pharmaceutical Technology, Deraya University, Minia 61519, Egypt; mahmoud.omar@deraya.edu.eg; 4Department of Pharmaceutics and Clinical Pharmacy, Faculty of Pharmacy, Sohag University, Sohag 82524, Egypt; 5Pharmacology and Toxicology Department, College of Pharmacy, Umm Al-Qura University, Makkah 21955, Saudi Arabia; 6Oral Medicine, Periodontology, Oral Diagnosis and Radiology Department, Ain Shams University, Cairo 11566, Egypt; mohamed.elmofty@nub.edu.eg; 7Oral Medicine, Periodontology, Oral Diagnosis and Radiology Department, Nahda University, Beni-Suef 62764, Egypt; 8Pharmacology and Toxicology Department, Faculty of Pharmacy, Nahda University, Beni-Suef 62764, Egypt; mohammed.gamal@nub.edu.eg; 9Department of Pharmacology and Toxicology, College of Pharmacy, Almaaqal University, Basrah 61014, Iraq; 10Department of Internal Medicine, Faculty of Medicine, Cairo University, Cairo 11562, Egypt; amel.salem@kasralainy.edu.eg; 11Department of Anatomy, Faculty of Medicine, Cairo University, Cairo 11562, Egypt; tarek.ibrahim@kasralainy.edu.eg

**Keywords:** nanovesicles, chitosan, mucin, irradiation, hepatic carcinoma

## Abstract

Liver cancer is a prevalent form of carcinoma worldwide. A novel chitosan-coated optimized formulation capped with irradiated silver nanoparticles (INops) was fabricated to boost the anti-malignant impact of rosuvastatin calcium (RC). Using a 2^3^-factorial design, eight formulations were produced using the solvent evaporation process. The formulations were characterized in vitro to identify the optimal formulation (Nop). The FTIR spectra showed that the fingerprint region is not superimposed with that of the drug; DSC thermal analysis depicted a negligible peak shift; and XRPD diffractograms revealed the disappearance of the typical drug peaks. Nop had an entrapment efficiency percent (EE%) of 86.2%, a polydispersity index (PDI) of 0.254, a zeta potential (ZP) of −35.3 mV, and a drug release after 12 h (Q12) of 55.6%. The chitosan-coated optimized formulation (CS.Nop) showed significant mucoadhesive strength that was 1.7-fold greater than Nop. Physical stability analysis of CS.Nop revealed negligible alterations in VS, ZP, PDI, and drug retention (DR) at 4 °C. The irradiated chitosan-coated optimized formulation capped with silver nanoparticles (INops) revealed the highest inhibition effect on carcinoma cells (97.12%) compared to the chitosan-coated optimized formulation (CS.Nop; 81.64) and chitosan-coated optimized formulation capped with silver nanoparticles (CS.Nop.AgNPs; 92.41). The bioavailability of CS-Nop was 4.95-fold greater than RC, with a residence time of about twice the free drug. CS.Nop has displayed a strong in vitro–in vivo correlation with R^2^ 0.9887. The authors could propose that novel INop could serve as an advanced platform to improve oral bioavailability and enhance hepatic carcinoma recovery.

## 1. Introduction

Long-term statin use has been shown in recent oncology trials to lower the incidence of ovarian, colon, breast, pancreatic, and hepatic tumors, among other cancers [1]. Statins are frequently used to decrease cholesterol.

Medications are known to slow or halt plaque formation and reduce long-term mortality in patients with cardiovascular disease. Notably, numerous studies have shown that statins play a crucial role in inhibiting the growth of cancer cells by regulating the mevalonate pathway [2,3]. Statins were initially believed to be contraindicated for patients with liver problems, which limited their usage in these patients when transaminase levels were found. Later research, however, has demonstrated that statins are safe and useful in treating some hepatic diseases, such as non-alcoholic fatty liver disease (NAFLD), cirrhosis, and hepatitis [4,5,6]. Statins may also reduce the incidence of liver cancer, according to encouraging studies [7]. Statin use has increased among patients with hepatocellular carcinoma (HCC), despite the lack of evidence that statins lower the incidence of HCC [8,9]. Compared to other statins, rosuvastatin calcium (RC), a strong inhibitor of HMG-CoA reductase, has the highest affinity for the enzyme’s active site and has been demonstrated to have better hepatic absorption in rats [10]. Furthermore, there is strong evidence that RC shields cells from ischemia damage [11,12,13].

Studies have also demonstrated that large quantities of lipophilic medications can be encapsulated in nanosuspension (NS) vesicles with an oil core. When encapsulated in suitably sized NS systems, these vesicles promote the transport and uptake of RC by cancer cells due to their improved permeability, retention effect, and sustained release properties [14,15,16,17,18].

Nanovesicles (NVs) have attracted substantial attention for their capacity to increase the bioavailability of active medicinal substances compared to conventional dose forms. Because of their unique structure, NVs, such as nonionic surfactants with or without cholesterol (CH), can encapsulate molecules that are both lipophilic and hydrophilic [19]. By preventing oxidation, NVs improve drug stability, prolong drug release, and improve drug penetration, which makes them highly compatible with biological systems [20,21,22]. Numerous uses are made possible by the increased stability of NVs, including improving cancer treatment [23,24] and increasing bioadhesive qualities [24].

It is interesting to note that surface modification by the addition of positive charges can improve system stability, facilitate effective drug distribution, and get around the drawbacks of conventional formulations. By interacting with the mucin in the gastrointestinal (GIT) membrane, medication delivery vehicles with a positive charge can regulate drug release and improve absorption [25]. Applications of nanovesicles in gene therapy agents, drug delivery systems, antiviral, antifungal, antibacterial, anticancer, biosensors, and diagnostics for chronic illnesses are becoming more and more common [26,27,28,29,30,31,32]. The potent antibacterial properties of silver nanoparticles (AgNPs), which work against a range of drug-resistant bacteria, have also been emphasized by recent studies [27]. Additionally, AgNPs have become a major focus of radiation treatment research. Research has demonstrated that, both in vitro and in vivo, silver nanoparticles improve the radiosensitivity of glioma cells. Furthermore, studies have shown that radiation therapy using silver nanoparticles is more effective than radiation therapy alone for hepatocellular carcinoma, gastric cancer, and breast cancer [33].

Polymer-coated metallic nanoparticles have garnered significant attention due to their unique physicochemical properties and extensive range of potential applications. Chitosan-coated silver and gold nanoparticles are one such promising bioactive hybrid material; they have high biological activity, low toxicity, biocompatibility, and are biodegradable [34]. A biopolymer known as chitosan (CS) is considered non-toxic and has shown exceptional antifungal and antibacterial infection [35,36]. Both Gram-positive and Gram-negative bacteria have been demonstrated to have their growth inhibited by CS [37,38,39]. The development of metallic nanoparticles stabilized by CS creates new opportunities for stable and highly bioactive medicinal agents. Moreover, CS facilitates surface alteration via electrostatic interactions and enhanced surface absorption, which increases the therapeutic potential of the nanoparticle [40,41,42,43].

Until now, no approaches have examined X-ray-irradiated nanovesicular systems specifically targeting hepatic cells. Therefore, this novel study focuses on three key objectives: first, to fabricate nanovesicles using a factorial design (2^3^) to identify the optimal formulation for enhancing RC absorption; second, to use a chitosan coating to improve drug absorption and prolong RC retention in the stomach through mucoadhesion; and third, to investigate the use of this formulation, INop, for treating hepatic carcinoma.

## 2. Materials and Methods

### 2.1. Materials

Rosuvastatin calcium (RC), a potent HMG-CoA reductase inhibitor, was a kind gift from (Egyptian International Pharmaceutical Industries Co., Ciba Cairo, Egypt). The surfactants Span 20 (S20) and Span 60 (S60) were purchased from (Al-Nasr Chemicals Company, Cairo, Egypt). We bought cholesterol (CH) from Acros Organics in Cairo, Egypt. The organic solvents, methanol and chloroform, were of analytical grade and were purchased from (El-Nasr Pharmaceutical Chemicals Co., Cairo, Egypt). Phosphate-buffered saline (PBS) at pH 7.4 was prepared using sodium chloride, potassium chloride, sodium phosphate, and potassium phosphate, purchased from (Sigma Aldrich, Cairo, Egypt). For the dissolution studies, the paddle-type dissolution apparatus (Erweka DT-720, Langen, Germany) was used, and the release medium consisted of PBS (pH 7.4), prepared as per standard protocols. For the spectrophotometric analysis, a Shimadzu UV-2401 PC spectrophotometer (Kyoto, Japan) was utilized to measure absorbance at a wavelength of 241 nm. The cellophane membrane (4.5 cm^2^) used in the dissolution tests was sourced from (SERVA Electrophoresis GmbH, Heidelberg, Germany), and was soaked in the release medium for 24 h before use. Chitosan (CS) (degree of deacetylation ~75%) with low molecular weight was (kindly provided by Amoun Pharmaceutical Company, Al Obour, Cairo, Egypt), and was used for coating the nanovesicles. Every reagent was used exactly as supplied, requiring no additional purification, and all glassware was thoroughly cleaned and sterilized before use.

### 2.2. Experimental Design

Using the Minneapolis, USA-based Design-Expert^®^ 10 program (version 10.0.6.0), eight formulations (N1–N8) with a 2^3^ factorial design were created to examine the effects of three independent variables on the characteristics of nanovesicles (NVs). The concentration of cholesterol (X3), the kind of surfactant (X2), and the surfactant concentration (X1) were the independent variables. The vesicle size (VS, Y1), polydispersity index (PDI, Y2), entrapment efficiency (EE%, Y3), zeta potential (ZP, Y4), and drug release after 12 h (Q12, Y5) [44] were the five dependent variables taken into consideration. Table 1 provides a summary of the experimental design. (checked with Section 3.1).

### 2.3. Preparation of RC-Loaded Nanovesicles (RC. NVs)

To produce a stable suspension of RC-loaded nanovesicles, the vesicles were made using a modified solvent evaporation approach [45]. A 1:1 methanol–chloroform combination was used to dissolve the necessary amounts of RC (80 mg), Span 20 or Cholesterol, and Span 60 (in a 2:1 or 3:1 molar ratio). The organic solvents were selected because they effectively dissolved the surfactants and the lipophilic drug (RC). A magnetic stirrer (IKA, Germany) was used to continuously agitate the solution at 60 °C to evaporate the organic solvents. A thin lipid layer comprising the medicine and surfactants formed on the flask walls as a result of this procedure. Phosphate-buffered saline (PBS, pH 7.4) was then used to hydrate the thin film [46], which assisted in rehydrating the lipids and forming nanovesicles. Because the hydration procedure was conducted at room temperature, the lipid film had time to expand and create the nanovesicles that contained the RC. Following the formation of the vesicles, the suspension underwent several filtration processes to guarantee a uniform vesicle size and eliminate any bigger particles. For stability testing, the finished suspension was kept in storage at 4 °C. This method was chosen due to its convenience of use and effectiveness in creating nanovesicles with regulated drug release characteristics and high encapsulation efficiency.

Based on their capacity to stabilize nanovesicles and improve the encapsulation of lipophilic medications such as rosuvastatin calcium (RC), the surfactants Span 20 and Span 60 were chosen. While Span 60 offers superior vesicle stability and encapsulation efficiency, Span 20, with a lower HLB, is best for stabilizing lipophilic RC. In order to decrease permeability and stop aggregation, cholesterol was added to the nanovesicles to improve membrane stability and rigidity. The 80 mg dosage of RC was selected to provide a consistent and regulated release profile while satisfying standard therapeutic requirements and guaranteeing effective encapsulation without overburdening the formulation.

### 2.4. Characterization of RC. NVs

#### 2.4.1. EE%

The suspension of nanovesicles was prepared, and the mixture was centrifuged for three hours at 15,000× *g* (using [SIGMA 3–30 K, Darmstadt, Germany]) in order to pellet the nanovesicles. After lysing the pellets with methanol and diluting the solution with PBS (pH 7.4), the supernatant which contains the free drug was meticulously collected, and the drug concentration in the supernatant was measured using a UV–visible spectrophotometer (Shimadzu UV-1800, Tokyo, Japan) at 241 nm to estimate the entrapped RC [47]. By deducting the free drug in the supernatant from the total amount of drug employed in the formulation, the amount of encapsulated RC was calculated.

The following formula was then used to determine the EE%: [45]
(1)$$EE\%=\frac{Quantity\;of\;drup\;trapped}{Total\;quantity\;of\;drug\;in\;the\;formulation}*100$$

#### 2.4.2. Evaluation of VS, PDI, and ZP

The obtained NVs were tested for VS, PDI, and ZP using a dynamic light scattering zeta-sizer (Malvern, Worcestershire, UK). To obtain the desired scattering intensity, certain amounts of formulations were combined with a specific amount of filtered water. At 25 °C room temperature, triplicates of each sample were utilized. The polydispersity index (PDI) was used to assess the distribution of vesicle sizes [48].

#### 2.4.3. Release Study of RC from Designed Formulations (Q12)

The paddle-type dissolution device [49] (device II, Erweka DT-720, Germany) was used to assess the in vitro release of rosuvastatin calcium (RC) from the developed nanovesicle (NV) formulations. The purpose of the release profile was to be assessed by the release study of RC from the formulations over 12 h while simulating the gastrointestinal environment. A modified dissolving procedure was used for this investigation. Before use, test tubes were immersed in the release medium (PBS, pH 7.4) for 24 h. One end of the tubes was sealed with a 4.5 cm^2^ cellophane membrane. These test tubes took the place of the paddles in the dissolution apparatus after being firmly knotted with cotton threads. The device was configured to run at 100 rpm, and the release medium was 500 mL of PBS at pH 7.4 that was kept at a steady 32 ± 0.5 °C to replicate the temperature of the human body [50,51]. In addition, 3 mg of RC and lyophilized NV formulations with similar concentrations of RC were employed for the release research. To maintain sink conditions, 3 mL of the release medium was removed and replaced with an equivalent volume of fresh PBS at predefined time intervals (0, 1, 1.5, 2, 2.5, 3, 4, 5, 8, and 12 h). The spectrophotometer was a Shimadzu UV-2401 PC (Kyoto, Japan), which was used to perform spectrophotometric analysis on the withdrawn samples at 241 nm. A linear regression equation based on the calibration curve was applied to determine the proportion of RC dissolved at every time point. To identify the release mechanism, the release profiles of RC from various formulations were examined, and the drug release kinetics were assessed using the appropriate mathematical models such as the Korsmeyer–Peppas, Higuchi, zero-order, and first-order.

#### 2.4.4. Drug Release Kinetics

The created release data of RC from the NV formulations were fitted using statistical kinetic models, including the Higuchi, first-order, and zero-order models [52,53]. The model with the highest correlation coefficient (R2) was chosen to simulate the kinetics of drug release. Furthermore, the Korsmeyer–Peppas Equation (2) was employed to describe the drug release mechanism [54].
(2)$$\frac{Mt}{M\infty }={Kt}^{n}$$

The drug diffusion coefficient, or (n), is the slope of the curve that illustrates the relationship between log Mt/M∞ and log *t*. In this case, t is the release time, k is the kinetic constant, and Mt/M∞ is the drug release percentage.

### 2.5. Selection of Nop

The choice of Nop [36] was based on the highest desirability from statistical analysis using Design Expert 10 software.

### 2.6. Preparation of CS.Nop

The Nop was coated with CS (0.2 mg/mL). The chitosan was subjected to sonication (Sonix IV USA, SS101H230, Springfield, VA, USA) after dissolving in a 0.1% *v*/*v* acetic acid solution. A magnetic stirrer operating at 500 rpm was used to gradually incorporate 10 mL of CS solution into the NV suspension. The resultant suspension was left for two hours on the stirrer [45].

### 2.7. Characterization of CS.Nop

#### 2.7.1. Transmission Electron Microscopy (TEM)

Nop and CS.Nop were morphologically investigated using transmission electron microscopy (Eindhoven, The Netherlands) to determine how the CS coat impacts the morphology of the created Nop. We diluted the sample with distilled water to prepare it for measurement. In this experiment, we used filter paper to remove any excess material after letting a droplet of the sample rest on a copper grid covered with carbon for 90 s at 25 °C. We used phosphotungstic acid to negatively stain the vesicles before the evaluation and allowed them to dry. Photomicrographs were obtained at an adequate magnification, and the material was examined at reasonable intervals [55,56].

#### 2.7.2. FTIR Analysis

The FTIR spectra of RC, CH, S 60, and CS, along with their lyophilized Nop and CS.Nop, were recorded using FTIR (Waltham, MA, USA) at 4000–400 cm^−1^ [57].

#### 2.7.3. DSC Analysis

DSC (DSC 50 Shimadzu, Kyoto, Japan) was used to record the thermograms of RC, CH, S 60, and CS, along with their lyophilized Nop and CS. Throughout 20 to 400 °C, 5 mg of samples was heated in a conventional aluminum pan at a rate of 5 °C/min, with a constant flow of inert nitrogen 25 mL/min [58,59].

#### 2.7.4. XRPD Analysis

The identical content utilized in the DSC investigation was subjected to XRPD analysis. Diffractograms (PW 1140, Columbus, OH, USA) and Cu-k radiation were used to record the XRPD patterns. Diffractograms were generated at a 2°/mm scanning speed and 2°/2 cm per 2 charts [60].

#### 2.7.5. Mucoadhesive Strength

Mucoadhesive efficacy was measured by evaluating the amount of mucin that adhered to the selected Nop and CS.Nop. Fifty milligrams of mucin was dissolved in one hundred milliliters of PBS (pH 7.4) to create a mucin dispersion before the experiment. The dispersion was then left spinning at 500 rpm on a stirrer (Hei Dolph, Chicago, IL, USA) overnight. After combining the two mixtures, they were shaken for two hours in a thermostatically regulated environment at 37 °C in a shaking water bath (GFL 3033, Burladingen , Germany) containing a 0.5 mg/mL mucin solution in PBS (pH 7.4) [45]. The blends were then centrifuged for one hour at 11,000× *g*. After collecting the supernatants, an ultraviolet–visible double-beam spectrophotometer (made in the US by Labomed Inc., Los Angeles, CA, USA) was used to measure the concentration of free mucin at 595 nm. The mucin concentration was ascertained by utilizing the calibration curve’s regression Equation (3), as follows:(3)y=0.0023X+0.0052 

The mucoadhesive strength was calculated as a mucin adsorption % from Equation (4)
(4)$$Mucin\;adsorption\;\%=\frac{Mt-Mf}{Mt}*10$$

(Mt) is the total quantity of mucin used and (Mf) is the free quantity of mucin in dispersion [61].

#### 2.7.6. In Vitro Drug Release

Since the Nop and CS.Nop will be administered orally, the in vitro dissolution experiments must be conducted in media of different pH to represent the stomach (pH 1.2), duodenum (pH 5.4), and intestine (pH 7.4). The release study of RC from Nop and CS.Nop was carried out as mentioned before at a wavelength of 241 nm.

#### 2.7.7. Drug Release Kinetics

Nop and CS.Nop were examined kinetically using the Higuchi diffusion model, zero-order kinetics, and first-order kinetics.

#### 2.7.8. Physical Stability Study

The physical stability of CS.Nop was assessed at 25 ± 2 °C, 4 ± 1 °C, and 37 ± 2 °C. Freshly made CS.Nop was kept for three months and tested for VS, PDI, ZP, and drug retention percent (DR%) at the beginning and end of each month.

### 2.8. Synthesis of Ag NPs

The **Phyllanthus Emblica** (Indian gooseberry) plant extract was obtained from the Department of Pharmacology, Cairo University’s Faculty of Pharmacy, and the plant material was collected from the medicinal plant farm for the Faculty of Pharmacy, Cairo University. The plant was carefully washed, dried, and then powdered. The extract was prepared by soaking the powdered plant material in distilled water for 48 h, followed by filtration to obtain a clear extract. This extract was then used to synthesize silver nanoparticles (AgNPs). Next, 10 mL extract and a 1 mM AgNO_3_ (90 mL) solution were combined and forcefully swirled in an Erlenmeyer flask to reduce the silver ions. The solution’s color changes from brown to dark brown. We spun the mixture in a centrifuge at 80,005× *g* for fifteen minutes to remove the unwanted organic material. It was then dried in an oven after being rinsed twice with deionized water.

Using UV–Vis spectroscopy, the production of AgNPs was verified. A distinctive absorption peak at 415 nm, which correlates to the surface plasmon resonance (SPR) of the AgNPs, demonstrated the reduction of silver ions to nanoparticles. It is commonly acknowledged that the production of silver nanoparticles is characterized by this spectral characteristic. The updated publication will include the UV–Vis spectrum to demonstrate the creation of nanoparticles [62].

#### Characterization of AgNPs

A zeta sizer was used to determine the PZ, PDI, and ZP (Santa Barbara, CA, USA).

### 2.9. Preparation of Nop. AgNPs and CS.Nop. AgNPs

To create the capped formulation (Nop. AgNPs), we modified the aforementioned method by hydrating the Nop with 10 mL of colloidal AgNPs instead of PBS during the preparation of various NV formulations. CS 0.2% solution was added dropwise, stirring with a magnet stirrer, to create the capped optimized chitosan-coated formulation (CS.Nop. AgNPs) [45,47].

### 2.10. Preparation of Irradiated Formulations

Both AgNPs and CS.Nop. AgNPs were exposed to X-ray for 10 min. For three successive days, the two formulations received 200 cGy/min beams of 6 MV X-rays produced by a linear accelerator (2D Panoramic—Orthopantomogram Machine, France). The irradiated samples were wrapped in lead foils until they were used (Appendix A).

### 2.11. Cytotoxicity Study

To enhance cell growth, 50% gentamycin/mL and 10% inactivated fetal calf serum were added to the RPMI-1640 medium. Two or three times a week, the cells were kept in a humidified environment at 37 °C with 5% CO_2_. Corning^®^ 96-well tissue culture plates were used for all anti-cancer experiments. After being seeded at a density of 5 × 10^4^ cells/well, the tumor cell lines were cultured for a full day. RC, CS.Nop, AgNPs, AgirNPs, CS.Nop.AgNPs, and the irradiation-optimized formulation (INop) were among the twelve concentrations of each drug that were examined. Each concentration had three replicates and ranged from 1, 2, 3.9, 7.8, 15.6, 31.25, 62.5, 125, 250 to 500 µg/mL. Each 96-well plate had six vehicle controls, which were the medium devoid of the active ingredient.

Cell viability was evaluated using the MTT test following a 48 h incubation period. After removing the culture medium from the 96-well plates, 100 µL of brand-new RPMI-1640 medium was added. All wells, including the untreated controls, received 10 µL of a 12 mM MTT stock solution (5 mg of MTT in 1 mL of PBS) after phenol red was eliminated. For four hours, the plates were incubated at 37 °C with 5% CO_2_. Following incubation, each well received 50 µL of dimethyl sulfoxide (DMSO) after the medium was withdrawn. A pipette was used to completely mix the wells, and they were then incubated for ten more minutes at 37 °C. A microplate reader (Sunrise, TECAN, Inc., Chapel Hill, NC, USA) was used to measure the optical density (OD) at 590 nm.

Cell viability was calculated by dividing the mean OD of wells treated with the test samples by the mean OD of untreated cells, and then multiplying the result by 100%. The relationship between drug concentration and cell survival was used to create a survival curve for each tumor cell line. GraphPad Prism software,10.4.1. (San Diego, CA, USA) [63] was used to generate the dose–response curve to estimate the 50% inhibitory concentration (IC50), or the concentration at which half of the cells become toxic. The six-vehicle controls for each 96-well plate were used to assess the effects of the solvents or dispersion agents used for the test compounds as well as the medium (RPMI-1640 with 10% fetal calf serum). The components of the vehicle controls were as follows:**Control 1 (Negative Control):** This contained neither nanovesicles (NVs) nor silver nanoparticles (AgNPs), but rather the same solvent that was used to dissolve the active substances (such as ethanol, PBS, or water). This made sure that any biological effects that were seen were not caused by the solvent itself.**Control 2–6:** These controls comprised 10% fetal calf serum in RPMI-1640 medium with the same amounts of vehicle solvents (such as ethanol and PBS) used to dissolve the test chemicals as in the experimental wells. These controls made sure that any cytotoxicity was attributable to the drug or formulation and helped assess any possible effects from the medium or vehicle alone.

### 2.12. In Vivo Study

The investigation used three groups of healthy male New Zealand rabbits with an average weight of 3.5 kg. Nop and CS.Nop formulations were given orally by gavage to groups I and II, while group III was given the drug suspension. Based on their body surface area, the equivalent amount of 0.2 mg/kg was administered to the rabbits [64].

Based on ethical issues and recommendations for reducing the number of animals used in experiments while still generating useful data, we employed three rabbits per group in our investigation. To lessen the possible influence on animal welfare and to adhere to the 3Rs principle (Replacement, Reduction, Refinement), which seeks to strike a compromise between ethical duty and scientific rigor, a small group size was selected.

A combination of previously published research and pharmacological considerations led to the dosage of 0.2 mg/kg. This dosage was specifically chosen to guarantee therapeutic relevance while preserving safety within the bounds set for comparable formulations. A foundation for choosing a safe and efficient dose range was established by earlier research with rosuvastatin calcium (RC) or comparable formulations in animal models. It has been observed that doses between 0.1 and 0.5 mg/kg produce therapeutic results without posing a serious risk of harm [65,66]. The dose was scaled for rabbits using interspecies allometric scaling, which considers metabolic differences between species. This ensures the dose is appropriate for rabbits and can be extrapolated to human therapeutic ranges. Moreover, initial trials were conducted to confirm that 0.2 mg/kg was effective and well-tolerated in the animal model, ensuring that the selected dose achieved the desired therapeutic outcomes without adverse effects.

The rabbits received the NV formulations and unrestricted access to water after an overnight fast. The ethics council of NAHDA University Faculty of Pharmacy granted ethical approval for the study’s methodology (Acceptance No. NUB-017-023, approval date: 13 June 2023). After the rabbits were given the formulations orally, blood samples were drawn from their ear veins into pre-heparinized glass tubes at specific intervals (0.5, 1.5, 2, 2.5, 3, 4,5, 6, 8, and 12 h). The collected blood samples were spun in a centrifuge at 4000× *g* for 10 min at 4 °C. Once transferred to 5 mL plastic tubes, the plasma was kept frozen at −70 °C until the analysis was complete. A week-long wash-out phase was used for a cross-over design.

The plasma samples were supplemented with atorvastatin (50 µL–100 ng/mL) to serve as an internal standard. Using 4 mL of ethyl acetate, the RC and atorvastatin were separated. Lab Sciex’s LC-MS/MS system (API-4000, Foster, CA, USA) was used to assess the drug, and the method’s accuracy, linearity, and lower limit of quantification were all verified. The mobile phase was a 4:1 *v*/*v* mixture of acetonitrile and 0.1% formic acid in water. The Agilent, CA, USA-based Zobrax Eclipse Plus column has dimensions of 4.6 × 50 mm and a particle size of 5 µm. With an isocratic flow rate of 0.9 mL/min, the injection volume was 15 μL [67]. The pharmacokinetic characteristics of RC following the oral administration of the NV formulations were examined using a PK-solver.

### 2.13. In Vivo–In Vitro Correlation

Plotting the percentage of drug released in vitro against the percentage of drug absorbed from the CS.Nop formulation at predetermined time intervals (0.5, 1, 1.5, 2, 2.5, 3, 4, 5, 6, 7, 8, and 12 h) allowed for the evaluation of the Nop formulation’s in vitro–in vivo correlation (IVIVC). Equation (5) was used to determine the in vivo proportion of medication absorption using the Wagner–Nelson method:(5)$$Percentage\;drug\;absorbed=\frac{Ct+Kel\left(AUC0-t\right)}{Kel\left(AUC0-\infty \right)}*100$$
where AUC0−t is the area under the curve from zero to time t, AUC0−∞ is the area under the curve from zero to infinity, Kel is the elimination rate constant, and Ct is the drug’s plasma concentration at time t [45].

Goodness-of-fit metrics were used to evaluate the IVIVC’s validity. The coefficient of determination (R2) was used to assess the degree of the linear relationship between in vitro release and in vivo absorption [63].

The IVIVC model’s validity is reinforced by this statistical study, which offers solid proof of its suitability for forecasting in vivo drug absorption from in vitro release data. By investigating other statistical indicators and external validation datasets, further research will improve the correlation.

## 3. Results and Discussion

### 3.1. Experimental Design

Design-Expert^®^10 software was used to statistically examine the factorial design data. Experimental tests on the viability of creating RC led to the selection of three factors: the amount of cholesterol, the type of surfactant, and the amount of surfactant. NVs and the two levels of each factor were set to guarantee that the signal-to-noise ratio was determined with precision, and that the model could be used to navigate the design space [66,67]. The ratio for each response (VS, PDI, EE%, ZP, and Q12) was greater than 4, which is ideal. For all parameters, the difference between the adjusted and projected R^2^ was less than 0.2, which is a fair level of agreement [66,67]. The results are illustrated in (Table 1).

### 3.2. Characterization of RC.NVs

The characterization parameters of the prepared NVs are listed in (Table 2).

#### 3.2.1. Drug Entrapment Efficiency % (EE%)

The EE% values of the generated NV formulations were limited to 61.4 ± 0.734 to 92.4 ± 0.941. The efficacy of RC encapsulation is strongly influenced by the ratio of CH to surfactant type. As demonstrated in N1 and N2, there is a considerable (*p* < 0.0068) increase in EE% up to a certain point when the CH ratio rises. According to reports, the stability and hydrophobicity of NVs increase when the CH ratio rises. Nevertheless, as observed in N4 and N6 [68,69,70], raising the CH ratio may cause the drug to be excluded and consequently result in a drop in EE% as RC molecules compete with it for available packing spaces [71].

Interestingly, it should be noted that EE% has shown a strong relationship with the type of surfactant used (*p* < 0.0022). Compared to their analogs made with S20, the NVs formed with S60 displayed a higher EE%. This might be explained by the fact that S 60 had a higher phase transition temperature than S20, a lower HLB value (4.7), and a longer chain structure [72]. A close look at N1 has shown the greatest EE (92.4 ± 0.941%) for NVs synthesized with S60. For S20, the N8 formulation has displayed the lowest EE (61.4 ± 0.734) due to the comparatively higher hydrophilicity (HLB 8.6) compared to S60.

ANOVA analysis has shown that the model F is significant (*p* = 0.0068). The final equation for the coded factors is as follows:(6)EE=+78.89+3.68A−8.61B+2.36C

#### 3.2.2. Measurements of VS, PDI, and ZP

The two most crucial elements in describing the prepared NVs are vs. and PDI. With the ratio of CH [73], the vesicle size of drug-loaded NVs grew dramatically (*p* = 0.0058). Additionally, it was demonstrated that the HLB value of the surfactant used had a substantial impact on the size of NVs; NVs created by S60 were smaller than those prepared by S20. This may be explained by S60, which made the vesicles more hydrophobic, which reduces surface-free energy and results in the formation of smaller vesicles. Furthermore, like in the case of S20 [74], the hydrophilicity of surfactants (higher HLB value) increases the water uptake of these nanovesicles, increasing their sizes.

The prepared NVs had PDI values that ranged from 0.278 ± 0.32 to 0.510 ± 0.05, which are deemed acceptable [75]. The very homogenous distribution of the produced vesicles could explain the low PDI values [76]. The equations of both VS and PDI are as follows:(7)VS=+24.91A+74.14B−8.19C
(8)PDI=+0.38+0.029A+0.088B−0.010C 

The net charge on the NV surface was measured via the zeta potential (ZP). The force that pushes the vesicles apart decreases as the charge on the NV surface does. As a result, the NV suspension becomes unstable and the vesicles adhere to one another. The results have revealed that the ZP values of NVs ranged from −43.7 ± 3.65 to −31.5 ± 1.92 mV, indicating the excellent physical stability of the prepared NVs. The ZP value of approximately ±30 mV often signifies system stability because of the electrostatic repulsion between vesicles [77].

The type of surfactant had a significant effect (*p* = 0.001) on the ZP of NVs. S20 displayed higher ZP values than those obtained with S60. This may be ascribed to the higher hydrophilicity of S20 compared to S60 [78].

#### 3.2.3. Release Study of Designed Formulations

The drug release patterns from the RC suspension and other NV formulations are illustrated in (Figure 1a,b). The dissolution pattern of RC has shown fast drug dissolution, reaching 100% after 3 h. This is because the medication is very soluble in alkaline pH [47]. When compared to the medication [79], other formulations have demonstrated a delayed release of RC. For N1, N2, N3, N4, N5, N6, N7, and N8, the total percentage of drug release throughout the first three hours was roughly 28.9%, 22.9%, 51.7%, 44.6%, 31.1%, 40.8%, 29.1%, and 56.5%, respectively.

The dissolution of the adsorbed drug at or just below the NV surface [80] caused an initial burst release in all formulations during the experiment. This was followed by a delayed release caused by the trapped RC within the specially constructed NVs. ANOVA analysis showed that the independent factors significantly affected drug release (*p* < 0.05). It is clear from Table 2 that the type of surfactant significantly (*p* = 0.0034) affected the amount of drug release. S20 has a significant increase in drug release compared to S60. This may be attributed to the higher HLB value of S20 [74].

As illustrated in Table 3, ANOVA and residual analysis confirmed the significance of the model used where an adequate precision of 10.05 was observed with close conformity between the adjusted R^2^ (0.8658) and the predicted R^2^ (0.6933), where the difference was less than 0.2. Following a polynomial study of the cumulative percent of RC released from NVs after 12 h, a linear model was chosen because, according to the statistical analysis, it could clearly show the major influence of independent variables on the RC release. Based on the results of an ANOVA data analysis, the following equation was generated.
(9)$$\%\;RC\;\left(Q12\right)=+63.08-5.55A+11.48B-1.5C\;$$

The 3D plot of the effect of CH concentration and surfactant type on VS, PDI, EE, ZP, and Q12 is represented in (Figure 2a–e).

#### 3.2.4. Kinetics of Release Study

To illustrate the release mechanism, various release kinetic models were used to fit the release patterns of the developed NV formulations. Table 4 displays the R^2^ values for various kinetic models. Formulations (1)–(7) followed an anomalous diffusion-controlled mechanism, also known as the Higuchi model, based on the highest R^2^ value. Using one-dimensional diffusion over the bilayer membrane, the RC was extracted from the mixtures. This model assumes that the drug’s solubility is greater than its original concentration in the matrix, while its diffusivity remains constant. Therefore, the release environment maintains the sink condition [81]. Additionally, the fact that the (n) values in the Korsmeyer–Peppas model were limited to values between 0.56 and 0.81 provided more proof that the primary means of RC release were diffusion and erosion, which do not involve Fickian transport [54]. The release of RC from formulation N8 followed first-order kinetics, indicating that the rate of release is concentration-dependent. This could be explained by looking at the nanosystem, where the surface changes because the gatekeeper disassembles due to the rapid hydrolysis of RC at pH 7.4. A small amount of the drug is introduced into the solution gradually during this procedure, forming a gradient from which the transport into the bulk solution takes place [82].

#### 3.2.5. Determination of the Nop

The determination of the Nop is based on the selection of the formulation that depicts the highest desirability. From the statistical analysis of different NV formulations using Design-Expert 10, it was found that the Nop formulation showed a desirability of 0.71. The composition of Nop is significantly (*p* < 0.05) different from the composition of all designed formulations; therefore, the Nop was prepared and characterized. Table 5 illustrates the composition and responses of Nop, and its experimental results (Nexp).

### 3.3. Characterization of Nop and CS.Nop

#### 3.3.1. Transmission Electron Microscopy (TEM)

Figure 3a,b depict transmission electron micrographs of Nop and CS.Nop. Nop has revealed that the vesicles formed were spherical and distributed uniformly in nanometric size, free of aggregation. CS.Nop has displayed a significant increase in size compared to Nop. A coat with a largely faded tint surrounds the core. These results demonstrate the successful coating of Nop with CS.

#### 3.3.2. FTIR Analysis

FTIR spectroscopy assesses the drug’s compatibility with the NV ingredients. Figure 4a–f show the spectra of different ingredients and formulations. The FTIR spectrum of RC revealed notable bands that corresponded to its functional groups. A broad band was seen at 3427.9 cm^−1^ due to O-H stretching. Another band was observed at 2923.56 cm^−1^, which is characteristic of the olefinic C–H of the heptanoic side chain. In the fingerprint region, many sharp bands are located at different wave numbers that reflect the physical properties of the drug; bands from 1120 to 1160 cm^−1^ indicate S=O stretching.

The prominent bands at 3396.83 cm^−1^ in the FTIR spectrum of CH were attributed to O-H stretching. The band at 2929.83 cm^−1^ is caused by the symmetric stretching vibration of CH_2_, and the band at 1464.21 cm^−1^ is attributed to the CH_2_ and CH_3_’s asymmetric stretching vibration. Additionally, CH displayed a distinctive band at 1376.45 cm^−1^, which was associated with the bending vibrations of CH_2_ and CH_3_, and at 1054.45 cm^−1^, which was associated with the bending vibration of C-O [83].

The S60 spectra showed distinct bands at 1467.19, 1735.24, 2916.22, and 3384.37 cm^−1^, which, in turn, correspond to aliphatic O-H stretching, ester carbonyl stretching, C-H stretching, and O-H bending [84]. The ketonic group (C=O) and amide group (-NH_2_), which are commonly associated with water molecules, are responsible for the bands seen in the CS spectrum at 3423.9, 1658.67, and 1549.52 cm^−1^ [85].

The characteristic bands of RC vanished in the fingerprint region of CS.Nop, which confirms the coating of the drug with CS [45]. On the other hand, the fingerprint region of the drug is not superimposed to those of Nop, indicating a change in the physical properties of the drug.

#### 3.3.3. Differential Scanning Calorimetry (DSC)

The thermograms of RC, Nop, and CS.Nop are shown in Figure 5a,b. RC exhibits a clear melting endotherm at 231.8 °C, indicating its crystalline nature [86]. In the thermogram of the lyophilized Nop combination, only the melting endotherm of CH is visible, with no peaks corresponding to the other components, likely due to the diluting effect. The thermogram of the lyophilized CS.Nop mixture shows that the characteristics of CS and CH dominate over the drug’s character, confirming that RC is fully coated with CS. The DSC thermogram reveals endothermic peaks of S60 and CH at 58.59 °C and 148.2 °C, respectively. Additionally, CS shows an exothermic peak at 310.4 °C, indicating CS degradation, and an endothermic peak at 87.4 °C, corresponding to water loss [87]. In both Nop and CS.Nop, there is a negligible peak shift, which may be attributed to the formation of vesicles with bilayers resulting from the interaction of lipid components [76].

#### 3.3.4. XRPD

Figure 6a–f show the diffractograms of the RC (a), S60 (b), CH (c), CS (d), Nop (e), and CS.Nop (f). The diffractogram of RC depicts two sharp peaks at the two binary points (21°, 7.9) and (22.1, 11.2), the *x*-axis point represents 2 theta, and the *y*-axis value represents the intensity percent (counts). The characteristic peaks for S60 were visible at the points (18.6°, 117.9), (23.8°, 506.8). The CH diffractogram has shown distinct peaks at (17.4°, 37.9). The binary points of other components and formulations are annotated in the figures. The lyophilized Nop and CS.Nop showed several peaks at different 2-theta positions. It is clear from the diffractogram of Nop that there is a synergistic effect resulting from the combination of CH and S60 in increasing the intensity of the crystallinity of the drug; this may be ascribed to the dilution factor [45]. The disappearance of the typical RC peaks in the diffractogram of the lyophilized CS.Nop and Nop suggest that the RC is amorphized and trapped within the prepared NVs [45]. This confirms that the CS has completely coated the vesicles. The diffraction patterns of incident rays on the crystalline materials are governed by Bragg’s law; when an X-ray beam hits a crystal surface, Brag’s law states that the beam’s angle of incidence and scattering will be the same. Another condition under which constructive interference occurs is when the path difference, denoted as (d), is equal to (n) wavelengths, where (n) is a full number (Equation (10)).
(10)nλ=2dsinθ

(λ) is the wavelength of the X-ray, and d characterizes the spacing between atomic planes. (θ) is the diffraction beam angle and n is the order of diffraction, and it is an integer [88].

**Figure 6 pharmaceutics-17-00072-f006:**
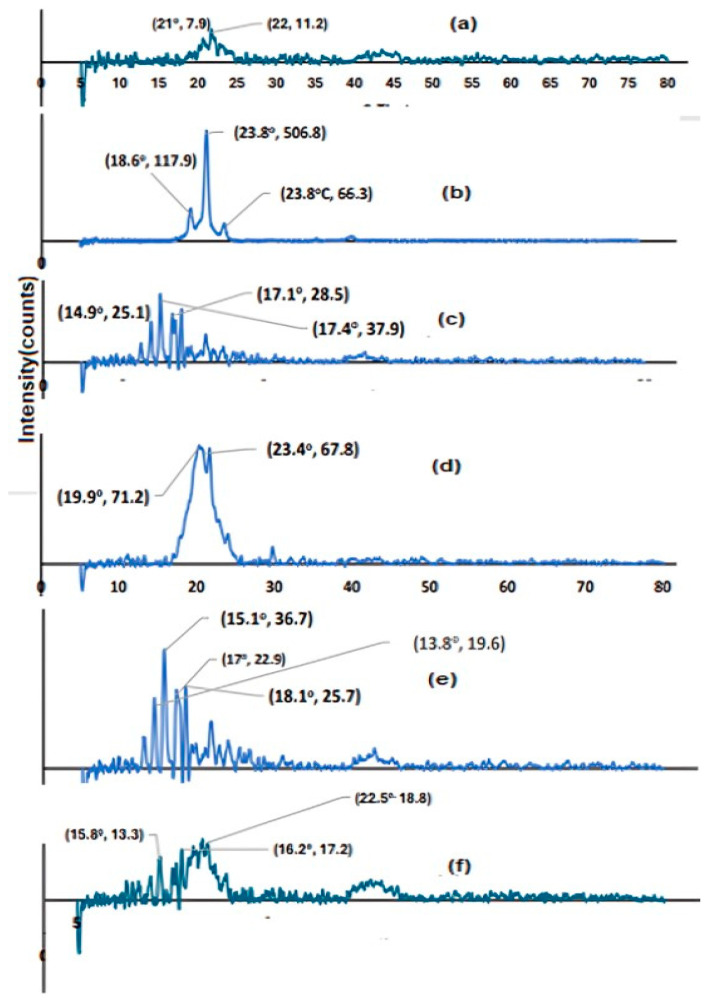
XRD of rosuvastatin calcium (RC, (**a**)), span 60 (S60, (**b**)), cholesterol (CH, (**c**)), chitosan (CS, (**d**)), optimized formulation (Nop, (**e**)), chitosan-coated optimized formulation (CS.Nop, (**f**)).

#### 3.3.5. In Vitro Drug Release from CS.Nop at Different pH

Figure 7a–c illustrate the drug release pattern of Nop and CS.Nop in different pH media when compared to the drug suspension. RC was released more slowly from Nop and CS.Nop formulations [47]. The insufficient solubility of the drug in an acid medium, the barrier effect of the NVs, and CS coatings may be the cause of the delayed release of the drug from the two formulations (Figure 7a). The high solubility of the drug in the higher pH media (5.4, and 7.4) contributed to the high drug release that was seen in both formulations (Nop and CS.Nop) (Figure 7b,c). The all-release media had a biphasic release pattern, with a rapid onset phase. The drug molecules adsorbed on the surface and the diffusion mechanism from an outer layer of vesicles are responsible for the quicker release [84]. The slow release of RC across the bilayer membrane resulted in a sustained release phase over the subsequent eight hours. The results have shown that Nop and CS.Nop had significantly different release characteristics (*p* < 0.05). Certainly, the in vitro release rate could not be regarded as a decisive reason for elucidating the bioavailability of the drug because it failed to demonstrate the CS coat’s in vivo mucoadhesive action. The data have shown that the release characteristics of the Nop and CS.Nop formulations differed significantly (*p* < 0.05).

The physicochemical characteristics of the nanovesicles (NVs) were largely determined by the surfactant selection. Span 20 improved drug encapsulation but may have compromised stability due to its shorter alkyl chain and lower hydrophilic–lipophilic balance (HLB), which produced smaller vesicles with less stiffness. Span 60, on the other hand, offered more vesicle rigidity and stability due to its longer alkyl chain and higher HLB, which helped to increase stability and control over drug release during storage. The zeta potential was also affected by the kind of surfactant; formulations of Span 60 exhibited stronger electrostatic stabilization [89]. Furthermore, cholesterol stabilized the membrane by adjusting the vesicle bilayer’s stiffness and fluidity. By decreasing membrane permeability and aggregation, higher cholesterol concentrations increased the stability of the nanovesicles [90]. This effect was demonstrated by longer drug release patterns, a lower polydispersity index (PDI), and reduced variability in vesicle size. The efficacy of medication encapsulation may be diminished by excessive cholesterol, although it may decrease the amount of space the drug has in the bilayer. Furthermore, by promoting strong electrostatic interactions with negatively charged mucin in the gastrointestinal tract, the positive charge of the chitosan (CS) coating significantly enhanced mucoadhesive properties. This coating also improved vesicle stability and extended release by creating an additional diffusion barrier that slowed drug release. The greater zeta potential values observed in CS-coated formulations further support its role in enhancing vesicle integrity and reducing aggregation [91].

#### 3.3.6. Kinetics of Drug Release

To illustrate the release mechanism, the release profiles of the Nop and CS.Nop formulations were fitted to several release kinetic models. Table 6 displays the R^2^ values for various kinetic models. Both formulations’ in vitro release at different pH media may have been illustrated by a diffusion-controlled mechanism [52,54].

The Korsmeyer–Peppas release model’s n values offer important insights into the fundamental mechanics of drug release, especially about diffusion and erosion processes. The following formula is commonly used to analyze medication release:M_t_/M_∞_ = kt^n^
where:M_t_ is the amount of drug released at time t;M_∞_ is the total amount of drug to be release;k is the release rate constant;n is the release exponent, which helps determine the mechanism of drug release.

Diffusion through the polymeric matrix is the primary driver of drug release when n = 0.5, as the release follows Fickian diffusion. The release occurs according to a case-II transport mechanism when n = 1, indicating that drug release is regulated by polymer erosion or swelling, with the polymer matrix breaking down and releasing the drug in a zero-order fashion. On the other hand, the release mechanism is usually categorized as anomalous (non-Fickian) diffusion when 0.5 < n < 1, where the drug release is facilitated by both diffusion and erosion. This is the most typical release pattern seen in controlled-release formulations, where the drug’s diffusion and the polymer matrix’s erosion both affect the drug’s rate of release.

Since the solubility and swelling behavior of the polymer matrix can alter with the pH, influencing both diffusion and erosion, the pH of the release media has a substantial effect on the release mechanisms. For instance, consider the following:

The polymer matrix may show less swelling and a slower erosion rate at an acidic pH (pH 1.2, imitating stomach circumstances), resulting in a more diffusion-controlled release (lower n values closer to 0.5). Here, Fickian diffusion mostly controls drug release. The polymer matrix is more prone to swell and break down at a neutral or basic pH (pH 7.4, mimicking intestinal circumstances), which can lead to anomalous diffusion or erosion-controlled release (higher n values closer to 1). This suggests that the release process involves both matrix degradation and diffusion.

In our study, by analyzing the ***n* values** under different pH conditions, we observed a shift in the release mechanism from **diffusion-controlled** at acidic pH to a more **erosion-controlled** or **combined mechanism** at a neutral or basic pH. This is consistent with the known behavior of polymeric systems, where the degree of polymer swelling and the solubility of the drug are influenced by the pH of the medium [92,93].

#### 3.3.7. Mucoadhesive Strength

One interesting method for improving the variable bioavailability of rosuvastatin calcium (RC) in nanovesicle (NV) formulations and prolonging its stomach retention is the chitosan (CS) coating. An analysis of the mucoadhesive characteristics of the optimized formulation (Nop) and chitosan-coated Nop (CS.Nop) revealed a notable improvement in mucin binding for the CS.Nop formulation, as illustrated in Figure 8. In particular, CS.Nop’s binding efficacy was 1.7 times higher than that of the uncoated Nop. This enhancement is ascribed to the electrostatic interactions that occur between the negatively charged mucin proteins in the gastrointestinal system and the positively charged amino groups of chitosan [45].

Although the chitosan layer significantly increased these effects, other processes like hydrophobic contacts and hydrogen bonding probably also played a role in the mucin binding seen in uncoated Nop, in addition to electrostatic attraction. Additionally, the CS coating gives the formulation a pH-responsive quality that allows for improved drug release in the stomach’s acidic environment and sustained release as the pH rises along the gastrointestinal tract. For RC, a drug with pH-dependent solubility, this property is very helpful because it guarantees a steady and regulated release of the drug, improving absorption and therapeutic sustainability.

The results are consistent with previous research on bioadhesive systems, which consistently shows that positively charged carriers offer greater gastrointestinal retention and targeted administration. Nevertheless, the study’s findings indicate that CS.Nop’s mucoadhesive strength is superior to that of comparable chitosan-coated systems, indicating a possible improvement in the formulation procedure or interaction processes.

There are still restrictions even though the improved mucoadhesion is encouraging. Mucoadhesive strength was mainly assessed in vitro in this study, which might not accurately reflect the dynamic circumstances of the gut environment in vivo. Future studies ought to examine in vivo mucoadhesion and how it affects the pharmacokinetics and therapeutic results of RC. Applying this strategy to additional medications with low bioavailability could confirm the adaptability and efficiency of the CS-coated NV platform.

The increased mucoadhesion of CS.Nop may be ascribed to hydrogen bonding and hydrophobic interactions between the chitosan coating and mucin, which would further reinforce its capacity to stick to the mucosal surface [94]. In addition to extending the nanovesicles’ retention at the absorption site, these contacts also aid in preventing premature drug release, enabling RC to be released gradually and under control, especially in the small intestine, where absorption is most effective.

Thus, CS.Nop formulations’ mucoadhesive qualities offer several advantages: they prolong the gastrointestinal residence duration, increase the likelihood of RC absorption, and lower drug loss from first-pass metabolism. The overall bioavailability is greatly improved by this enhanced retention in conjunction with the gradual, regulated release of RC. Additionally, RC’s pH-dependent solubility guarantees that CS.Nop formulations can effectively release the medication for the best absorption in the small intestine’s more basic environment [45]. The enhanced therapeutic results and bioavailability of RC in CS-coated nanovesicle formulations are largely due to the synergy between mucoadhesion and controlled release [95,96].

#### 3.3.8. Physical Stability Study

A stability study is a crucial step in evaluating any developed formulation, as it ensures the maintenance of key properties such as vesicle size (VS), zeta potential (ZP), and drug retention (DR) over time. In this study, the stability of the CS.Nop formulation was assessed over three months at three different storage conditions: room temperature (25 °C), 37 °C, and refrigerated temperature (4 °C). The results, as shown in Table 7, indicate that there were no statistically significant changes (*p* > 0.05) in VS, the polydispersity index (PDI), ZP, or DR% when stored at 4 °C, as confirmed by ANOVA analysis.

Since no appreciable changes in important parameters were noted, the results imply that keeping CS.Nop at refrigeration temperature preserves its physical stability. This is consistent with the common understanding that vesicular systems tend to be more stable at lower temperatures, which lessens the possibility of fusion or aggregation [97]. The vesicles’ inherent propensity to fuse and aggregate, which is a common problem with nanovesicular formulations held under less controlled settings, may be the cause of the increase in vesicle size at ambient temperature and 37 °C.

The CS.Nop formulation, interestingly, demonstrated better stability at 4 °C than at ambient temperature (25 °C) and 37 °C. This is in line with earlier research that emphasizes how crucial lower temperatures are for maintaining the physical properties of nanovesicles [98]. The vesicles’ bioavailability and effectiveness are likely preserved by the reduced temperature, which also keeps them from going through unfavorable modifications like aggregation or disintegration. The stability study’s findings highlight how crucial ideal storage conditions are for nanovesicle compositions. Given that CS.Nop remains stable at refrigerated temperatures, this formulation appears to be suitable for long-term preservation without suffering a major reduction in therapeutic potential. Given that long-term storage and transportation circumstances can change, its stability at lower temperatures may be particularly advantageous for pharmaceutical applications. Concerns regarding possible aggregation and decreased effectiveness over time are raised by the reported rise in vesicle size at room temperature and higher temperatures.

To further improve the formulation’s stability under a wider range of circumstances, future research should look into additional stabilizing chemicals or alternate storage techniques such lyophilization. Furthermore, in vivo research will be required to verify that the higher stability seen at lower temperatures corresponds to better therapeutic results in clinical settings.

### 3.4. Characterization of AgNPs

The zeta potential can be used to determine how stable nanoparticles are in aqueous solutions by measuring the surface charge potential. AgNP zeta potential values were found to be −27.62 mV. It was claimed that the generated nanoparticles have negatively charged surfaces and the particle size measured was 63.8 nm Figure 9 [99].

### 3.5. Cytotoxicity Study

Before the in vivo investigation, an in vitro cytotoxicity study was performed to evaluate the formulations’ potential anticancer activity [100]. Free RC, CS.Nop, CS.Nop.AgNPs, AgNPs, AgirNPs, and INop were incubated for 24 h with carcinoma cells, and the viability of the cell was evaluated using the MTT assay. The results, summarized in Table 8, demonstrate that cell viability decreased with increasing RC concentrations (1–100 µg/mL). Since its half-maximal inhibitory concentration (IC50) was 100.63 µg/mL, free RC showed the maximum cytotoxicity. Previous studies have similarly reported RC’s cytotoxic effects on various cell lines [101]. With the lowest IC50 value of 33.33 µg/mL, INop notably showed the strongest anticancer activity. The combined effects of the irradiation of silver nanoparticles and better drug distribution via the chitosan-coated vesicles may be the cause of INop’s improved performance. The differing speeds at which cells absorb free versus encapsulated RC could be one reason for drug-induced cytotoxicity [102]. The hydrophobic properties of encapsulated RC probably limit its availability at the cellular level by impeding its passage through the lipid bilayer [103]. In increasing order of IC50 values, the formulations showed different levels of cytotoxicity: free RC (100.63 µg/mL), CS.Nop.AgNPs (97.9 µg/mL), AgNPs (120.52 µg/mL), CS.Nop (132.32 µg/mL), INop (33.33 µg/mL), and Agir.NPs (56.19 µg/mL). The release of RC from the chitosan surface during incubation or the radiosensitizing effects of irradiation AgNPs on the polymer matrix could be the cause of the decreased IC50 for INop that was found [104]. INop is a promising medication delivery method for addressing cancerous cells because of its low cytotoxicity and biocompatibility. The % suppression of cancer cells for each formulation is shown in Figure 10.

By creating a unique formulation that combines irradiated silver nanoparticles (AgNPs) and chitosan-coated nanovesicles to improve therapeutic efficacy against liver cancer, this study represents a significant leap in drug delivery systems. A dual-purpose strategy is introduced by the creative incorporation of irradiated AgNPs into the vesicular system; the chitosan coating enhances bioadhesion and controlled drug release, while the AgNPs function as strong radiosensitizers, producing reactive oxygen species (ROS) that specifically harm cancerous cells. Together, these elements improve drug absorption, retention, and targeting of cancer cells, providing a more potent option than traditional treatments. Mechanistically, the irradiated AgNPs contribute to improved cytotoxicity by causing oxidative stress and magnifying DNA damage in cancer cells, a behavior not ob-served with non-irradiated formulations. Chitosan coating increases gastrointestinal residence length and targeted delivery by providing a positively charged surface that interacts electrostatically with negatively charged mucin and cancer cell membranes. In order to overcome the problems of low solubility and quick clearance, the hydrophobic rosuvastatin calcium (RC) is further stabilized by encapsulation within the vesicular lipid bilayer. Site-specific action is made possible by this encapsulation’s regulated, pH-responsive drug release, which also reduces systemic side effects.

The results are consistent with previous research on delivery methods based on nanoparticles, which highlights how these systems can enhance drug solubility, stability, and bioavailability [93,94]. But compared to similar systems, the improved irradiation formulation’s (INop) measured IC50 values are significantly lower, highlighting the increased effectiveness of integrating irradiated AgNPs with a chitosan-coated nanovesicular substrate. Further supporting the promise of this strategy for clinical translation is the five-fold improvement in RC bioavailability seen with the chitosan-coated formulation (CS.Nop), which outperforms traditional statin formulations.

Notwithstanding these encouraging outcomes, several restrictions need to be noted. The intricacy of in vivo tumor microenvironments, where elements like immunological response and medication metabolism may affect results, is not adequately captured by the in vitro cytotoxicity study. Furthermore, more research is needed to guarantee the safety of irradiated AgNPs’ long-term biocompatibility. In order to confirm the improved formulation’s safety profile and therapeutic efficacy, future research should concentrate on in vivo trials. Its therapeutic applicability could be further expanded by extending this strategy to other hydrophobic medications or investigating combination treatments with supplementary anticancer medicines.

### 3.6. In Vivo Pharmacokinetic Study

The ICH criteria validated the precision, accuracy, selectivity/specificity, and linearity of the analysis method. The HPLC chromatogram of RC-spiked blank plasma showed that the retention time of the natural plasma components and the RC did not interfere with each other. The analytical method was selective, with a linear relationship (correlation coefficient (R^2^) = 0.9988) between the analysis method and the RC in the plasma calibration curve. Table 9 and Figure 11 display the data along with the derived pharmacokinetic parameters. When comparing the C_max_ of free RC with that of Nop and CS.Nop, it was found that both formulations significantly (*p* < 0.05) increased C_max_ (666.8, 579.6 ng/mL), respectively, whereas the free RC depicted a C_max_ of 372.61 ng/mL. Additionally, the bioavailability of Nop and CS.Nop was about 1.9 and 5 times, respectively, greater than the bioavailability of RC. CS.Nop has shown a longer t_1/2_ and delayed T_max_. The prepared CS.Nop can extend the release of RC, which was reflected by the longer MRT [105]. CS.Nop is a promising approach for enhancing RC oral bioavailability. The CS.Nop formulation may have a higher bioavailability due to the bigger surface area of nano-sized vesicles [106].The explanation for CS.Nop’s effectiveness compared to Nop is that the particles’ positively charged surfaces allowed them to form an electrical bond with the membrane of the negatively charged gastrointestinal tract [107,108]. Another mechanism by which CS may enhance RC oral absorption is by opening tight junctions, which facilitates the utilization of the paracellular transport route [109]. A possible approach to increase RC bioavailability is to cover the vesicles with CS.

### 3.7. In Vitro–In Vivo Correlation

Figure 12 demonstrates that the CS.Nop has displayed a strong in vitro–in vivo correlation, with an R^2^ value of 0.9887 at the 0.5, 1.5, 2, and 2.5 h time intervals. The average difference between actual and anticipated values was also measured using the root mean square error (RMSE). With a low RMSE of (0.0975), from the values of RMSE and R^2^, it is clear that a robust correlation between in vivo/in vivo obtained data exists.

## 4. Conclusions

This study highlights the significant potential of a novel chitosan-coated nanovesicle system capped with irradiated silver nanoparticles (INops) to enhance the therapeutic efficacy and bioavailability of rosuvastatin calcium (RC) for treating hepatic carcinoma. The innovative formulation combines the mucoadhesive and controlled-release properties of chitosan with the radiosensitizing capabilities of silver nanoparticles, achieving a multifaceted drug delivery approach. The nanovesicles (NVs) were successfully fabricated using a factorial design with Design-Expert 10, yielding an average vesicle size of 346.1 nm, an encapsulation efficiency (EE%) of 78.9%, and a zeta potential (ZP) of −37.9 mV. Coating the Nop vesicles with a 0.2% acidic chitosan solution resulted in an increased vesicle size of 529.7 nm, an EE% of 62.03%, and a ZP of 32.25 mV, while maintaining spherical morphology as confirmed by TEM. FTIR, DSC, and XRPD analyses demonstrated the compatibility of RC with formulation excipients and a solid–solid transition to a crystalline form, ensuring formulation stability.

Mechanistically, the chitosan coating enhanced gastrointestinal retention and mucosal interaction, significantly improving the bioavailability of RC by nearly fivefold compared to conventional formulations. The coating’s pH-responsive properties ensured targeted and sustained drug release while minimizing systemic side effects. Additionally, the mucoadhesive strength of CS.Nop vesicles was 1.7 times greater than that of uncoated Nop vesicles, and the formulation displayed remarkable stability at 4 °C, with minimal alterations in critical parameters such as vesicle size, the polydispersity index, and drug retention over three months. INop exhibited the highest cytotoxicity against carcinoma cells and the most pronounced inhibitory effect compared to other formulations, driven by the radiosensitizing effects of irradiated silver nanoparticles, which enhanced reactive oxygen species production and DNA damage in malignant cells.

These results underscore the promise of INop as an advanced therapeutic platform, offering enhanced drug absorption, retention, and site-specific targeting for liver cancer treatment. However, translating these findings into clinical applications necessitates further in vivo studies to evaluate the formulation’s efficacy and safety comprehensively, particularly within the complex dynamics of tumor microenvironments and long-term biocompatibility. This work establishes a robust foundation for advancing nanovesicular systems in oncology, paving the way for innovative strategies to address challenging malignancies and significantly improve therapeutic outcomes.

## Figures and Tables

**Figure 1 pharmaceutics-17-00072-f001:**
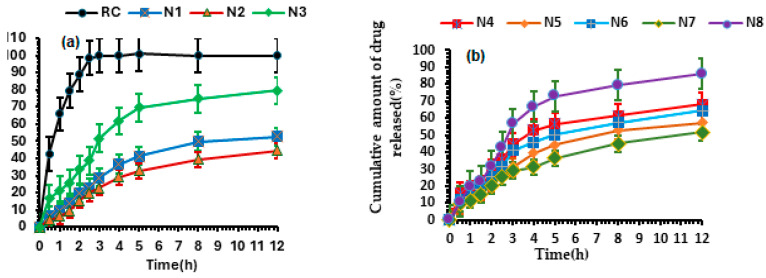
(**a**,**b**) Release profiles of the drug from different designed formulations in PBS pH 7.8 at 32 ± 0.5 °C.

**Figure 2 pharmaceutics-17-00072-f002:**
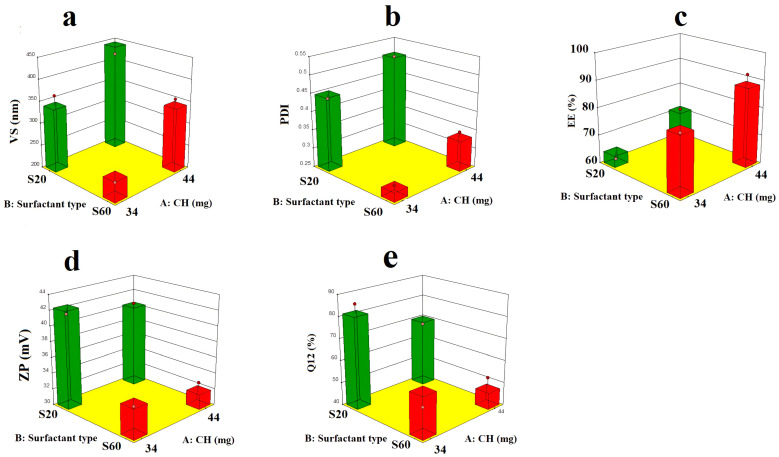
3D plot of the effect of cholesterol concentration and surfactant type on vesicle size (**a**), polydispersity index (**b**), entrapment efficiency (**c**), zeta potential (**d**), release after 12Q (**e**).

**Figure 3 pharmaceutics-17-00072-f003:**
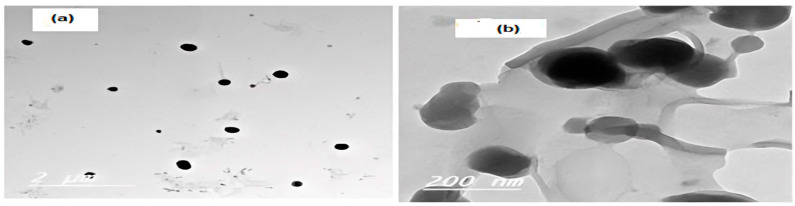
TEM of optimized formulation (Nop, (**a**)) and chitosan-coated optimized formulation (CS-No, (**b**)).

**Figure 4 pharmaceutics-17-00072-f004:**
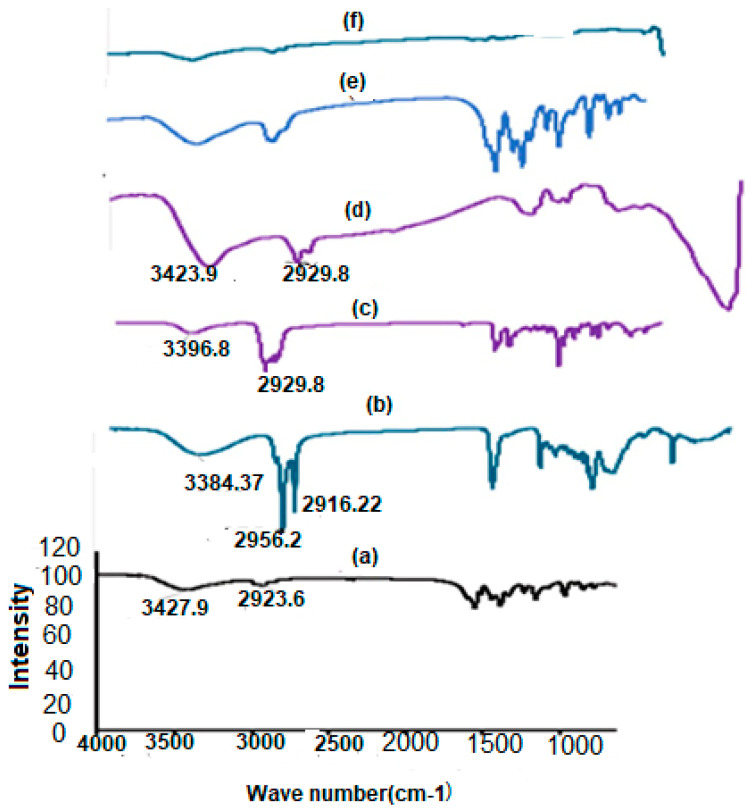
FTIR of rosuvastatin calcium (RC, (**a**)), span 60 (S60, (**b**)), cholesterol (CH, (**c**)), chitosan (CS, (**d**)), optimized formulation (Nop, (**e**)), chitosan-coated optimized formulation (CS.Nop, (**f**)).

**Figure 5 pharmaceutics-17-00072-f005:**
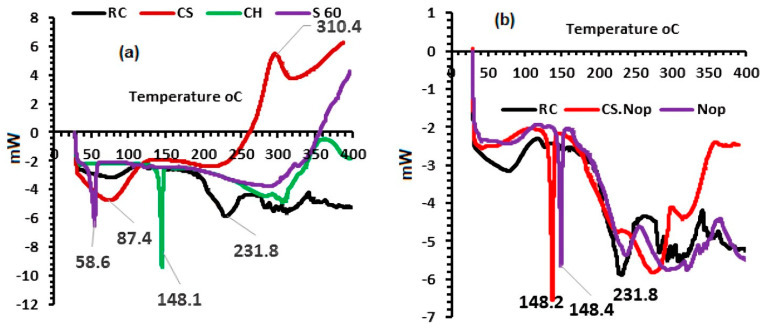
DSC of the following: (**a**): rosuvastatin calcium (RC), chitosan (CS), cholesterol (CH), span 60 (S60) and (**b**): RC, optimized formulation (Nop), chitosan-coated optimized formulation (CS.Nop).

**Figure 7 pharmaceutics-17-00072-f007:**
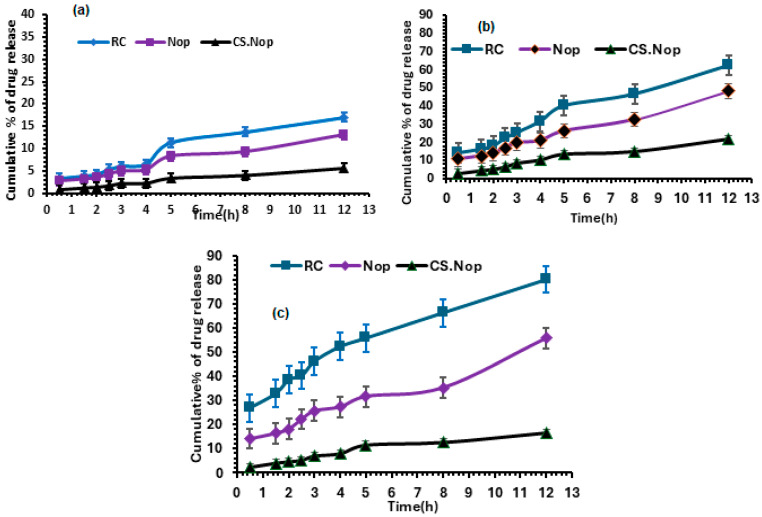
(**a**) Dissolution study of drug suspension and nanovesicle formulations in pH 1.2 RC, rosuvastatin calcium; Nop, optimized formulation; CS.Nop, optimized chitosan-coated formulation; (**b**) dissolution study of drug suspension and nanovesicle formulations in pH 5.4 RC, rosuvastatin calcium; Nop, optimized formulation; CS.Nop, optimized chitosan-coated formulation; (**c**) dissolution study of drug suspension and nanovesicle formulations in pH 7.4 RC, rosuvastatin calcium; Nop, optimized formulation; CS.Nop, optimized chitosan-coated formulation.

**Figure 8 pharmaceutics-17-00072-f008:**
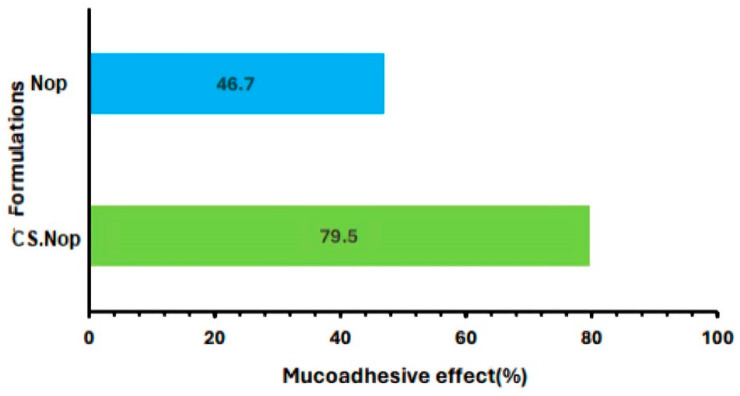
Mucoadhesive strength of optimized formulation (Nop) and chitosan-coated optimized formulation (CS.Nop).

**Figure 9 pharmaceutics-17-00072-f009:**
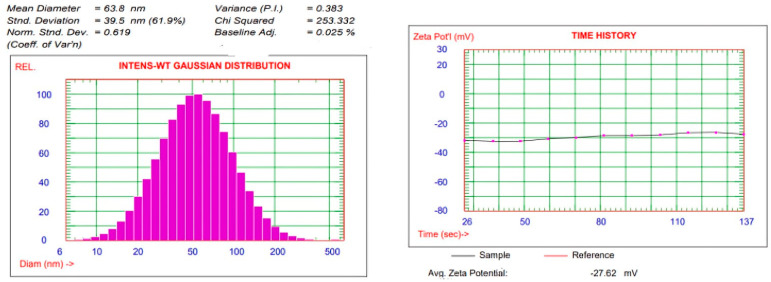
Particle size and zeta potential of silver nanoparticles.

**Figure 10 pharmaceutics-17-00072-f010:**
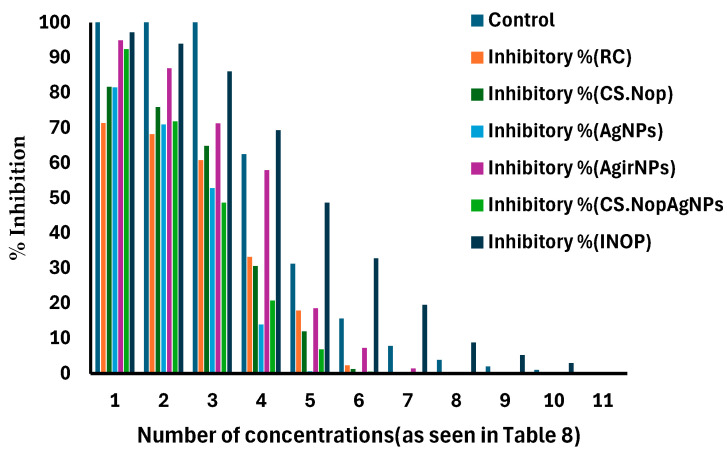
Inhibition pattern of carcinoma cells by different formulations: rosuvastatin calcium (RC); chitosan-coated optimized formulation (CS.Nop); silver nanoparticles (AgNPs); irradiated silver nanoparticles (AgiNPs); chitosan-coated optimized formulation.

**Figure 11 pharmaceutics-17-00072-f011:**
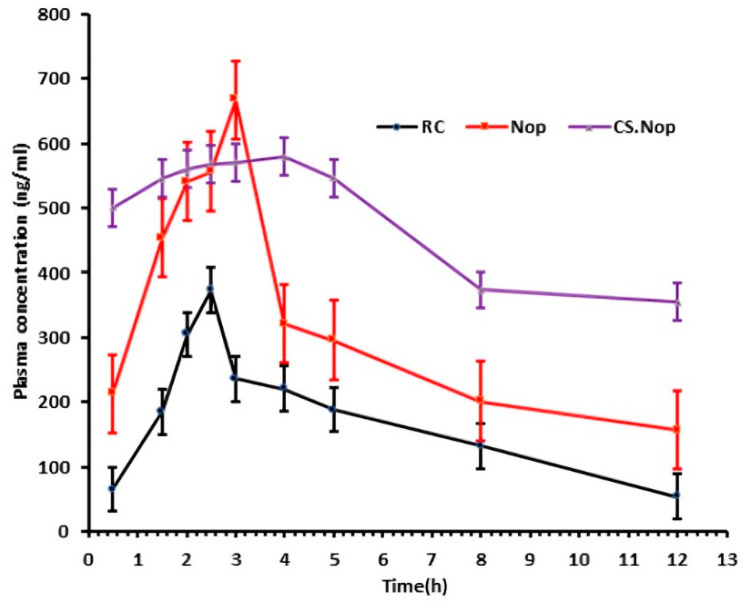
Time curve of plasma concentration following oral delivery of free rosuvastatin (RC), optimized formulation (Nop), and chitosan-coated optimized formulation (CS.Nop).

**Figure 12 pharmaceutics-17-00072-f012:**
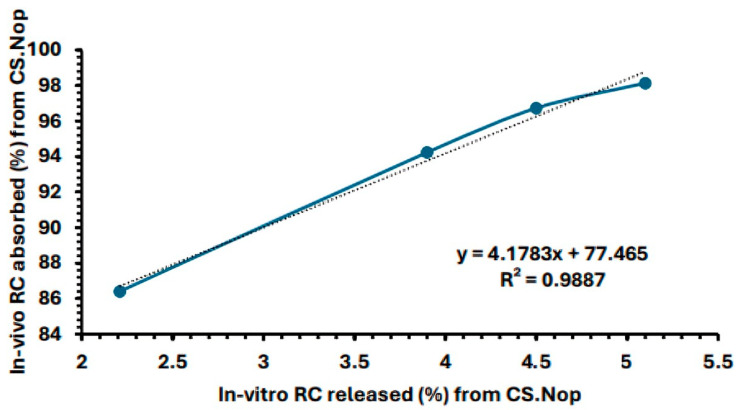
In vitro/in vivo correlation.

**Table 1 pharmaceutics-17-00072-t001:** Design of the experiment.

Factors Levels			
X1	Amounts of surfactant	88	100
X2	Type of surfactant	S20	S60
X3	Amount of cholesterol	34	44
Responses	Constraints		
Y1: Vs (nm)	Minimize		
Y2: PDI	Minimize		
Y3: EE%	Maximize		
Y4: ZP (mV)	Maximize		
Y5: Q_12_ (%)	Maximize		

Abbreviations: VS, vesicle size; PDI, polydispersity index, EE, entrapment efficiency, ZP, zeta potential, Q_12_, drug released after 12 h.

**Table 2 pharmaceutics-17-00072-t002:** Characterization of RC-NV formulations.

Formulation	CH (mg)	Surfactant	Amount of Surfactant (mg)	VS (nm)	PDI	EE%	ZP (mV) (−)	Q_12_
N1	44	S60	88	374.4 ± 6.23	0.347 ± 0.02	92.4 ± 0.941	−32.9 ± 2.13	52.7 ± 0.679
N2	44	S60	100	369.7 ± 7.25	0.284 ± 0.13	91.5 ± 1.641	−31.5 ± 1.92	44.7 ± 1.14
N3	34	S20	100	356.1 ± 11.46	0.427 ± 0.08	73.3 ± 0.768	−43.7 ± 3.65	79.6 ± 1.29
N4	44	S20	88	420.6 ± 8.35	0.505 ± 0.13	71.8 ± 0.210	−40.5 ± 4.21	68.2 ± 1.03
N5	34	S60	100	221.5 ± 12.66	0.266 ± 0.07	85.6 ± 1.82	−36.6 ± 2.91	57.2 ± 0.477
N6	44	S20	100	428.8 ± 18.31	0.510 ± 0.05	74.6 ± 0.463	−42.8 ± 1.08	64.5 ± 0.794
N7	34	S60	88	231.7 ± 13.61	0.278 ± 0.32	80.5 ± 0.874	−33.4 ± 2.15	51.8 ± 0.506
N8	34	S20	88	365.7 ± 15.71	0.439 ± 0.01	61.4 ± 0.734	−41.6 ± 1.06	85.9 ± 0.104

Note: Data are represented as mean ± SD (*n* = 3). Abbreviations: RC-NVs, rosuvastatin calcium loaded–nanovesicle suspension; VS, vesicle size; PDI, polydispersity index; EE%, entrapment efficiency; ZP, zeta potential; Q_12_, drug release after 12 h.

**Table 3 pharmaceutics-17-00072-t003:** Analysis of variance of responses and fit statistics of the factorial model.

Responses	Model	R^2^	Adjusted R^2^	Predicted R^2^	*p*-Value	F-Value	Adequate Precision
VS (nm)	Significant	0.9764	0.9587	0.9055	0.001	55.11	17.7
PDI	Significant	0.9752	0.9566	0.9009	0.0011	52.49	17.5
EE%	Significant	0.9392	0.8936	0.7569	0.0068	20.60	11.9
ZP (mV)	Significant	0.9393	0.8973	0.7571	0.0068	20.63	11.0
Q12 (%)	Significant	0.9233	0.8658	0.6933	0.0107	16.05	10.1

Notes: the difference between adjusted R^2^ and predicted R^2^ is less than 0.2, Adeq Precision measures the signal-to-noise ratio. A ratio greater than 4 is desirable. Abbreviations: VS, vesicle size; PDI, polydispersity index; EE, entrapment efficiency; ZP, zeta potential, Q_12_, drug release after 12 h.

**Table 4 pharmaceutics-17-00072-t004:** Kinetic analysis of the release data of RC from all designed nanovesicle formulations in PBS (pH 7.4).

Formulation	Zero-Order (R^2^)	First-Order (R^2^)	Higuchi (R^2^)	Mechanism	R^2^	n	Diffusion Type
N1	0.8427	0.8928	** 0.9477 **	Higuchi	0.9611	0.73	Anomalous diffusion
N2	0.8635	9098	** 0.9602 **	Higuchi	0.8653	0.81	Anomalous diffusion
N3	0.7866	0.8877	** 0.9053 **	Higuchi	0.9445	0.57	Anomalous diffusion
N4	0.7877	0.8727	** 0.924 **	Higuchi	0.9102	0.56	Anomalous diffusion
N5	0.8600	0.9156	** 0.9579 **	Higuchi	0.9720	0.64	Anomalous diffusion
N6	0.8254	0.9058	** 0.9325 **	Higuchi	0.9362	0.62	Anomalous diffusion
N7	0.9039	0.9505	** 0.9781 **	Higuchi	0.9698	0.57	Anomalous diffusion
N8	0.7675	** 0.9138 **	0.8959	First order	0.9325	0.71	First-order

**Notes:** R^2^ (coefficient of determination); n (diffusion coefficient according to Korsmeyer–Peppas model).

**Table 5 pharmaceutics-17-00072-t005:** Composition and responses of the Nop and Nexp.

Formulation	Composition		Responses				
	CH (mg)	S 60 (mg)	VS (nm)	PDI	EE%	ZP (mV)	Q_12_ (%)
Nop	34	100	211.43	0.254	86.2	−35.3	55.6
Nexp	34	100	223.2	1.000	84.8	−51.3	57.2

**Abbreviations:** CH, chitosan; S60, span 60; VS, vesicle size; PDI, polydispersity index; EE, entrapment efficiency; ZP, zeta potential, Nop; optimized formulation, Nexp; experimental formulation.

**Table 6 pharmaceutics-17-00072-t006:** Kinetic analysis of the release data of rosuvastatin calcium from predicted Nop and CS.Nop in different pH media.

pH	Formulation	Zero-Order (R^2^)	First-Order (R^2^)	Higuchi Model (R^2^)	Mechanism	R^2^	n	Diffusion Type
1.2	Nop	0.9149	0.9268	** 0.9540 **	Diffusion	0.9732	0.72	Anomalous diffusion
	CS.Nop	0.9132	0.9245	** 0.9524 **	Diffusion	0.9723	0.86	Anomalous diffusion
5.4	Nop	0.9113	0.9243	** 0.9379 **	Diffusion	0.9559	0.64	Anomalous diffusion
	CS.Nop	0.8992	0.9509	** 0.9648 **	Diffusion	0.9792	0.65	Anomalous diffusion
7.4	Nop	0.8440	0.9001	** 0.9232 **	Diffusion	0.9547	0.59	Anomalous diffusion
	CS.Nop	0.9666	0.9488	** 0.9666 **	Diffusion	0.9716	0.67	Anomalous diffusion

**Notes:** R^2^ (coefficient of determination; n (diffusion-coefficient according to Korsmeyer–Peppas model).

**Table 7 pharmaceutics-17-00072-t007:** Stability study of the CS.Nop nanosuspension at room temperature (25 °C ± 2 °C), refrigerated temperature (4 °C ± 1 °C), and 37 °C ± 2.

Storage Temp (°C)	Parameters		Storage Period (Month)		
		0	1	2	3
4 ± 1	VS	529.3 ± 801.2	441.3 ± 92.4	433.0 ± 112.5	431.6 ± 112
	PDI	1.000	0.528	0.664	0.891
	ZP	+25.6 ± 1.10	+24.11 ± 0.212	+25.71 ± 0.614	+24.23 ± 0.712
	DR%	99.91 ± 0.081	100.1 ± 0.00	99.9 ± 0.121	100 ± 0.00
25 ± 2	VS	529.3 ± 801.2	560 ± 203.8	586.7 ± 312.9	598.8 ± 538.6
	PDI	1.000	1.000	0.852	0.981
	ZP	+25.26 ± 1.10	+22.32 ± 0.961	+23.61 ± 0.563	+20.42 ± 0.851
	DR%	99.91 ± 0.081	99.64 ± 0.11 0	98.6 ± 1.02	97.9 ± 0.021
37 ± 2	VS	529.3 ± 801.2	491.3 ± 71.4	486.8 ± 59.3	4851.8 ± 59.3
	PDI	1.000	0.621	0.621	0.621
	ZP	+25.6 ± 1.10	+23.11 ± 0.212	+26.14 ± 0.316	+24.5 ± 0.612
	DR%	99.91 ± 0.081	99.6 ± 0.121	100.6 ± 0.043	99.4.6 ± 0.231

Notes: Data are represented as mean ± SD (n = 3). Abbreviations: CS.Nop, chitosan-coated optimized formulation; VS, vesicle size; PDI, poly dispersity index; ZP, zeta potential; DR%, drug retention%.

**Table 8 pharmaceutics-17-00072-t008:** Inhibitory activity of different formulations against hepatocellular carcinoma cells (means ± SD, n = 3).

**A**						
**Conc (µg/mL)**	**AgirNPs**		**CS.Nop.AgNPs**		**INop**	
	**Viability%**	**Inhibitory %**	**Viability%**	**Inhibitory %**	**Viability%**	**Inhibitory %**
500	5.12 ± 0.543	94.88 ± 0.456	7.59 ± 0.345	92.41 ± 0.675	2.81 ± 0.678	97.19 ± 0.567
250	13.06 ± 0.982	86.94 ± 0.895	28.14 ± 0.781	71.86 ± 0.567	6.04 ± 0.792	93.96 ± 0.345
125	28.74 ± 0.643	71.26 ± 0.346	51.36 ± 0.567	48.64 ± 0.452	13.96 ± 0.876	86.04 ± 0.378
62.5	42.05 ± 0.546	57.95 ± 0.671	79.28 ± 0.567	20.72 ± 0.668	30.65 ± 0.545	69.35 ± 0.872
31.25±	81.43 ± 0.324	18.57 ± 0.876	93.17 ± 0.981	6.83 ± 0.678	51.38 ± 0.681	48.62 ± 0.456
15.6	92.73 ± 0.567	7.27 ± 0.678	99.51 ± 0.675	0.49 ± 0.345	67.19 ± 0.456	32.81 ± 0.632
7.8	98.60 ± 0.998	1.4 ± 0.654	100 ± 0.871	0	80.48 ± 0.341	19.52 ± 0.467
3.9	100 ± 0.341	0	100 ± 0.786	0	91.23 ± 0.467	8.77 ± 0.467
2	100 ± 0.234	0	100 ± 0.341	0	94.76 ± 0.567	5.24 ± 0.346
1	100±	0	100 ± 0.678	0	97.02 ± 0.323	2.98 ± 0.567
0	100±	0	100 ± 0.321	0	100 ± 0.678	0
**B**						
**Conc (µg/mL)**	**RC**		**CS.Nop**		**Ag NPs**	
	**Viability%**	**Inhibitory %**	**Viability %**	**Inhibitory %**	**Viability %**	**Inhibitory %**
500	7.34 ± 0.98	71.32 ± 0.419	6.73 ± 0.392	81.64 ± 0.818	7.84 ± 0.891	81.45 ± 0.671
250	20.63 ± 0.871	68.20 ± 0.532	14.05 ± 0.456	75.95 ± 0.865	29.03 ± 0.872	70.97 ± 0.881
125	39.24 ± 0.945	60.76 ± 0.498	35.19 ± 0.613	64.81 ± 0.873	47.21 ± 0.912	52.79 ± 0.756
62.5	66.83 ± 0.756	33.17 ± 0.653	69.42 ± 0.543	30.58 ± 0.765	86.09 ± 0.599	13.91 ± 0.786
31.25	82.14 ± 0.786	17.86 ± 0.498	88.06 ± 0.435	11.94 ± 0.791	99.42 ± 0.897	0.58 ± 0.121
15.6	97.69 ± 0.891	2.31 ± 0.521	98.75 ± 0.321	1.25 ± 0.668	100 ± 0.654	0
7.8	100 ± 0.651	0	100 ± 0.439	0	100 ± 0.664	0
3.9	100 ± 0.678	0	100 ± 0.348	0	100 ± 0.785	0
2	100 ± 0.976	0	100 ± 0.621	0	100 ± 0.899	0
1	100 ± 0.564	0	100 ± 0.342	0	100 ± 0.921	0
0	100 ± 0.432	0	100 ± 0.543	0	100 ± 0.782	0

**Abbreviations:** RC, rosuvastatin calcium; CS.Nop, chitosan-coated optimized formulation; AgNPs, silver nanoparticles. Notes: IC50 (50–inhibitory concentration): 100.63, 97.96, 120.52 µg/mL for RC, CS.Nop, and AgNPs, respectively. **Abbreviations:** AgirNPs, irradiated silver nanoparticles; CS.Nop.AgNPs, chitosan-coated optimized formulation capped with silver nanoparticles; INop; irradiated optimized formulation. **Notes:** IC50 (50–inhibitory concentration): 56.19, 132.32, and 33.33 µg/mL for AgirNPs, CS.Nop.AgNPs, and INop, respectively.

**Table 9 pharmaceutics-17-00072-t009:** Pharmacokinetic parameters of free rosuvastatin calcium, optimized formulation (Nop), and chitosan-coated optimized formulation (CS.Nop) after oral administration to male Sprague Dawley rats.

Pharmacokinetic Parameters	RC	Nop	CS.Nop
Kel (h^−1^)	0.089 ± 0.621	0.093 ± 0.19	0.043 ± 0.039
Cl (L/h)	0.029 ± 0.091	0.015 ± 0.054	0.006 ± 0.061
t1/2 (h)	7.73 ± 0.731	8.47 ± 0.621	15.9 ± 0.921
C_max_ (ng/mL)	372.61 ± 39.211	666.8 ± 31.362	579.6 ± 29.620
AUC_0–12_ (ng·h/mL)	1857.94 ± 211.367	3428.48 ± 68.113	5617.89 ± 301.043
AUC_0–∞_ (ng·h/mL)	2785.22 ± 299.625	5211.04 ± 411.482	13,787.47 ± 811.362
AUMC_0–12_ (ng·h^2^/mL)	9309.95 ± 945.402	16,957.69 ± 441.523	17,619.27 ± 568.428
AUMC_0–∞_ (ng·h^2^/mL)	30,748.83 ± 1621.628	18,7317.79 ± 1436.612	303,680.8 ± 1915.202
MRT12 (h)	5.01 ± 0.682	4.95 ± 0.521	3.14 ± 0.914
MRT_0–∞_	11.04 ± 0.423	11.05 ± 0.325	22.03 ± 0.421

Notes: Data are represented as mean ± SD (n = 6).

## Data Availability

Data are contained within the article.

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
