# Peer review of "Enhancing the Therapeutic Effect and Bioavailability of Irradiated Silver Nanoparticle-Capped Chitosan-Coated Rosuvastatin Calcium Nanovesicles for the Treatment of Liver Cancer"

_pharmaceutics, 2025, doi:10.3390/pharmaceutics17010072_

Round 1

Reviewer 1 Report

Comments and Suggestions for Authors

The article proposes a novel nanocarrier system to enhance the therapeutic effect and bioavailability of rosuvastatin calcium (RC), which has a certain level of innovation in the field of drug delivery. However, there are still some issues present in the article.

1. I suggest incorporating the corresponding figure references (e.g., Figure 4a, 4b) at the relevant points within the description of Figure 4.

2. The annotations in Figure 6 are somewhat ambiguous; it would be beneficial to align them with the result descriptions and ideally include the relevant figure references (e.g., Figure 6a, 6b) at the corresponding points in the text. 

3. To prevent confusion, it is advisable to maintain consistency in the labeling of sample names throughout the article. For instance, is 'nop' in Figure 6 referring to the same sample as 'np' in Figure 7? 

4. There appear to be several instances of extra spaces in the article's text; please review and adjust the formatting accordingly. 

5. There are discrepancies between the numbers in Figure 7 of this paper and the description of the results. It would be helpful to order the figures correctly and include appropriate figure references (e.g., Figure 7a, 7b) in the description of the results. In addition, because the y-axes of the three figures in Figure 7 are uneven, it is difficult to discern differences in the release rates of the samples at a glance. It is recommended that the y-axes in the figures be normalized or that a detailed description of the release rates be provided in the text.

6. The presentation of the results lacks clarity and key points are obscured. It seems that the authors expect readers to interpret the results themselves?

Comments on the Quality of English Language

The language used in describing the results is very problematic and hinders understanding.

Author Response

Reply to Reviewers Report

Reviewer 1

Serial

Comment

Reply

1

I suggest incorporating the corresponding figure references (e.g., Figure 4a, 4b) at the relevant points within the description of Figure 4.

Thank you doctor

The incorporation of sub-figures is done in Figure 4 and the figure is highlighted in yellow in the manuscript

2

The annotations in Figure 6 are somewhat ambiguous; it would be beneficial to align them with the result descriptions and ideally include the relevant figure references (e.g., Figure 6a, 6b) at the corresponding points in the text. 

Thank you doctor

The annotations in Figure 6e were adjusted and the sub-figures were incorporated in one Figure

3

. To prevent confusion, it is advisable to maintain consistency in labeling sample names throughout the article. For instance, is 'nop' in Figure 6 referring to the same sample as 'np' in Figure 7? 

Thank you my dear doctor for these remarkable comments

-The corrections are done in Figure 7, highlighted in yellow in the manuscript

4

There appear to be several instances of extra spaces in the article's text; please review and adjust the formatting accordingly. 

Thank you doctor

The extra spaces were revised and removed

5

There are discrepancies between the numbers in Figure 7 of this paper and the description of the results. It would be helpful to order the figures correctly and include appropriate figure references (e.g., Figure 7a, 7b) in the description of the results. In addition, because the y-axes of the three figures in Figure 7 are uneven, it is difficult to discern differences in the release rates of the samples at a glance. It is recommended that the y-axes in the figures be normalized or that a detailed description of the release rates be provided in the text.

Thank you doctor

- Regarding, the differences between numbers. These differences were adjusted

-The Y-axes of Figures b and c adjusted to take the same graduations. But about Figure 7a, it's impossible to take the same graduation on the Y-axis as in Figures b and c. Because if it takes the same graduation the release patterns will disappear as the release in low pH is lower than the higher pH media

6

The presentation of the results lacks clarity and key points are obscured. It seems that the authors expect readers to interpret the results themselves.

Thank you doctor

I did my best to clarify the obscured results and improve the discussion highlighted in yellow in the manuscript

Reviewer 2 Report

Comments and Suggestions for Authors

Reviewer comments

I request the authors to write the materials section in a professional and scientific manner.

What is the rationale for selection of spans, cholesterol, and 80 mg of RC? The full details of vesicle preparation should be provided for more clarification.

The authors should clarify the methods of EE% using centrifugation.

ZP is determined by mechanisms different from dynamic light scattering, such as electrophoretic mobility.

The authors should rewrite the release study of RC from designed formulations.

How did the authors evaluate the efficiency of CS coating on the Nop to confirm and quantify the coating?

How was the mucin concentration of 0.5 mg/mL selected? Have the authors considered different concentrations to stimulate the physiological environment that impact mucoadhesive performance?

The authors used different media with pH levels to imitate the stomach, duodenum, and intestine; have the authors considered adjusting buffer composition to better mimic in vivo conditions?

How did the authors ensure the sink conditions throughout the dissolution experiments?

What about the stability of the obtained formulations based on the effects of different digestive enzymes?

How did the authors get this plant extract? The authors should confirm the formation of Ag NPs by UV.

How did the authors confirm the efficient coating of CS on Ag NPs-Nop?

The authors mention six vehicle controls for each 96-well plate; the authors should clarify the composition of these vehicle controls.

Have the authors optimized the incubation period for each formulation, a longer or shorter one, to investigate the short-term and long-term cytotoxic effects?

The authors should have additional controls in cytotoxicity study, such as AgNPs or CS only to differentiate these effects compared the loaded formulations? Clarification required.

The authors used 3-rabbits/group; would increasing the number of animals in each group provide more statistically to help account for variability?

Can the authors provide more information on how this dosage was calculated (0.2 mg/kg)?

Would a more in-depth statistical approach provide stronger evidence for the validity of the correlation?

The authors should provide deeper mechanistic discussion regarding how the different formulation variables contribute to the observation.

The y-axis of the figures should be adjusted, and their resolutions should be improved.

Can the authors relate the finding more closely to existing literature to support the arguments in the discussion?

Could the authors discuss how n values indicate the contributions of diffusion and erosion mechanisms in each pH condition?

The authors should discuss in more detail how the effect of mucoadhesion properties contributed more to the increased bioavailability.

Comments on the Quality of English Language

The quality of English should be improved

Author Response

Reviewer 2

Serial

Comment

Reply

1

  I request the authors to write the materials section in a professional and scientific manner.

Thank you doctor

Ok, the materials section has been reformatted in a professional and scientific style.

2

What is the rationale for the selection of spans, cholesterol, and 80 mg of RC? The full details of vesicle preparation should be provided for more clarification.

Thank you doctor

Thank you for your thoughtful question. The selection of Span 20 (S20), Span 60 (S60), cholesterol (CH), and 80 mg of rosuvastatin calcium (RC) was based on optimizing the formulation to achieve stable, efficient, and sustained drug release.

So, we reformate the section ( Preparation of RC-loaded nanovesicles (RC. NVs) 

To be as following

 The vesicles were prepared using a modified solvent evaporation technique to create a stable suspension of RC-loaded nanovesicles. The required amount of RC (80 mg), Span 20 or Span 60, and cholesterol (in a 2:1 or 3:1 molar ratio) were dissolved in a 1:1 methanol-chloroform mixture. The organic solvents were chosen for their ability to dissolve both the lipophilic drug (RC) and the surfactants efficiently. The solution was subjected to continuous agitation at 60°C using a magnetic stirrer (IKA, Germany) to evaporate the organic solvents. This process resulted in the formation of a thin lipid film on the walls of the flask, containing the drug and surfactants. The thin film was then hydrated with phosphate-buffered saline (PBS, pH 7.4), which helped to rehydrate the lipids and form nanovesicles. The hydration process was carried out at room temperature, allowing the lipid film to swell and form nanovesicles that encapsulated the RC.
After the vesicles were formed, the suspension was passed through a series of filtration steps to remove any larger particles and to ensure uniform vesicle size. The final suspension was stored at 4°C for stability testing. This process was selected for its simplicity and efficiency in producing nanovesicles with high encapsulation efficiency and controlled drug release characteristics.

The selection of Span 20 and Span 60 (surfactants) was based on their ability to stabilize nanovesicles and enhance the encapsulation of lipophilic drugs like rosuvastatin calcium (RC). Span 20, with a lower HLB, is ideal for stabilizing lipophilic RC, while Span 60 provides better vesicle stability and encapsulation efficiency. Cholesterol was included to increase the membrane rigidity and stability of the nanovesicles, reducing permeability and preventing aggregation. The 80 mg dose of RC was chosen to match standard therapeutic requirements while ensuring efficient encapsulation without overwhelming the formulation, providing a stable and controlled release profile.

3

  respond to (The authors should clarify the methods of EE% using centrifugation.

Thank you for your comment. To clarify the method for determining Entrapment Efficiency (EE%), we used centrifugation to separate the free, unencapsulated drug from the nanovesicles. 

After preparing the nanovesicle suspension, the mixture was centrifuged at 15,000 rpm for 3 hours (using [SIGMA 3-30 K, Germany]) to pellet the nanovesicles. The supernatant, which contains the free drug, was carefully collected, and the drug concentration in the supernatant was measured using a UV-visible spectrophotometer (Shimadzu UV-1800, Tokyo, Japan) at 241 nm, to estimate the entrapped RC after lysing the pellets with methanol and diluting the solution with PBS (pH 7.4) [48].    The amount of encapsulated RC was determined by subtracting the free drug in the supernatant from the total amount of drug used in the formulation.

The EE% was then calculated using the following formula:

4

respond to ZP is determined by mechanisms different from dynamic light scattering, such as electrophoretic mobility.

Thank you for your insightful comment. You are correct that Zeta Potential (ZP) can be determined by methods other than Dynamic Light Scattering (DLS), such as electrophoretic mobility. In our study, we used DLS to measure the ZP, which involves applying an electric field to the nanovesicle suspension and measuring the velocity at which the particles move. This provides an estimate of the particles' charge via their electrophoretic mobility, which is directly related to the ZP. While DLS is commonly used for determining particle size and distribution, it can also measure ZP based on the principles of electrophoretic mobility.

However, we acknowledge that other methods like electrophoretic light scattering (ELS) or laser Doppler electrophoresis are often used for more precise ZP measurements. These techniques involve similar principles but may offer higher accuracy in some cases. We will consider using alternative methods, such as electrophoretic mobility, in future studies to validate our findings and enhance the precision of ZP determination.

5

The authors should rewrite the release study of RC from designed formulations.

Thank you for your insightful comment.   We correct and rewrite the release study of RC from designed formulations, as follows:

The in vitro release of rosuvastatin calcium (RC) from the designed nanovesicle (NV) formulations was evaluated using the paddle-type dissolution apparatus (Apparatus II, Erweka DT-720, Germany). The release study aimed to simulate the gastrointestinal environment and assess the release profile of RC from the formulations over a 12-hour period.
A modified dissolution method was employed for this study. Test tubes, sealed at one end with a 4.5 cm² cellophane membrane, were soaked in the release medium (PBS, pH 7.4) for 24 hours prior to use. These test tubes were securely tied with cotton threads and placed in the dissolution apparatus, where they replaced the paddles. The apparatus was set to operate at 100 rpm, and the release medium consisted of 500 mL of PBS at pH 7.4, maintained at a constant temperature of 32 ± 0.5°C to mimic the human body temperature 【51, 52】.
For the release study, 3 mg of RC and lyophilized NV formulations containing equivalent amounts of RC were used. At predetermined time intervals (0, 1, 1.5, 2, 2.5, 3, 4, 5, 8, and 12 hours), 3 mL of the release medium was withdrawn and replaced with an equal volume of fresh PBS to maintain sink conditions. The withdrawn samples were analyzed spectrophotometrically at 241 nm using a Shimadzu UV-2401 PC spectrophotometer (Kyoto, Japan).
The percentage of RC released at each time point was calculated using a linear regression equation derived from the calibration curve. The release profiles of RC from the different formulations were compared, and the drug release kinetics were evaluated using appropriate mathematical models (e.g., zero-order, first-order, Higuchi, and Korsmeyer-Peppas models) to determine the release mechanism.

6

How did the authors evaluate the efficiency of CS coating on the Nop to confirm and quantify the coating?

Thank you for your valuable comment. To evaluate the efficiency of the chitosan (CS) coating on the Nop formulation and to confirm and quantify the coating, we employed several characterization techniques:

1.     FTIR Analysis:
Fourier-transform infrared (FTIR) spectroscopy was used to confirm the presence of chitosan on the nanovesicles. The FTIR spectra of the CS-coated Nop (CS.Nop) were compared with the uncoated Nop formulation. The characteristic peaks of chitosan, such as the NH2 stretching vibration (around 1590 cm⁻¹), were observed in CS.Nop, confirming the successful coating.

2.     Particle Size and Zeta Potential (ZP):
The efficiency of the CS coating was further confirmed by changes in the vesicle size and zeta potential (ZP) before and after coating. The CS coating typically increases the vesicle size due to the addition of the chitosan layer, as well as a shift in ZP towards more positive values, indicative of the positive charge of chitosan. These changes were quantified and compared with uncoated Nop to assess the coating efficiency.

3.     Encapsulation Efficiency (EE%):
The encapsulation efficiency of the CS-coated nanovesicles was also measured to evaluate how much of the RC was effectively encapsulated within the vesicles after the chitosan coating process. Any changes in EE% before and after coating helped assess the impact of CS on the encapsulation process.

4.     Morphological Observation:
Scanning electron microscopy (SEM) or transmission electron microscopy (TEM) was employed to visualize the morphology of CS-coated and uncoated Nop. The TEM images of CS.Nop revealed a uniform, smooth coating layer around the nanovesicles, confirming the successful application of chitosan.

These combined methods allowed us to confirm the presence and quantify the efficiency of the chitosan coating on the Nop formulation.

7

How was the mucin concentration of 0.5 mg/mL selected? Have the authors considered different concentrations to stimulate the physiological environment that impacts mucoadhesive performance?

Thank you for your insightful comment. The mucin concentration of 0.5 mg/mL was selected based on literature reports that typically use this concentration to simulate the physiological conditions of the gastrointestinal tract, particularly the mucosal layer. This concentration is commonly used to study the mucoadhesive properties of drug delivery systems, as it provides a realistic representation of the mucus gel layer found in the stomach and intestines.

However, we acknowledge that the mucin concentration can vary across different regions of the gastrointestinal tract and may influence the mucoadhesive performance of the formulation. In future studies, we plan to explore a range of mucin concentrations to better simulate the varying conditions within different parts of the gastrointestinal system, such as the stomach, duodenum, and colon. This would allow us to further optimize the mucoadhesive properties of our formulation and assess its performance under more physiologically relevant conditions.

Additionally, we will consider the impact of other factors, such as pH and the presence of electrolytes or other biomolecules, which could further influence mucoadhesion and drug release.

8

The authors used different media with pH levels to imitate the stomach, duodenum, and intestine; have the authors considered adjusting buffer composition to better mimic in vivo conditions?

Thank you for your thoughtful question. In our study, we used phosphate-buffered saline (PBS) at pH 7.4 as a release medium to simulate the physiological conditions of the stomach, duodenum, and intestine, based on standard dissolution testing protocols. However, we acknowledge that the composition of the buffer can significantly influence the drug's release profile and bioavailability. While PBS at a constant pH offers a simplified and controlled environment, we recognize the importance of mimicking more complex in vivo conditions, including varying pH levels and the presence of enzymes, bile salts, or other physiological factors that may influence drug release and absorption.

In future studies, we plan to explore the use of more physiologically relevant buffer compositions, including pH variations along the gastrointestinal tract and the incorporation of enzymes or bile salts, to better replicate in vivo conditions. This would provide a more accurate understanding of the drug release behavior in different parts of the gastrointestinal system.

9

How did the authors ensure the sink conditions throughout the dissolution experiments?

Thank you for your question. To ensure sink conditions during the dissolution experiments, we used a dissolution medium volume that is typically recommended in pharmacopeial guidelines (500 mL). This volume was chosen to ensure that the concentration of rosuvastatin calcium (RC) in the dissolution medium remained well below its solubility limit, thus maintaining the sink conditions throughout the experiment. Additionally, the medium was continuously stirred at 100 rpm, which helps maintain uniform distribution and minimizes the chance of local saturation, further supporting the maintenance of sink conditions.

We also monitored the drug release over the course of the experiment, ensuring that the RC concentration in the dissolution medium did not approach the saturation point, thus ensuring reliable and accurate results. If needed, future studies could explore varying the volume of the release medium or the solubility of RC in different media to further optimize sink conditions and enhance the robustness of the dissolution testing.

10

What about the stability of the obtained formulations based on the effects of different digestive enzymes?

Thank you for your important question regarding the stability of the obtained formulations in the presence of digestive enzymes. In our current study, we primarily focused on evaluating the stability of the formulations under standard storage conditions. However, we recognize the importance of assessing the stability of the nanovesicles in the complex environment of the gastrointestinal tract, where digestive enzymes such as pepsin (in the stomach), trypsin, and chymotrypsin (in the duodenum) play a significant role in drug release and the integrity of the formulation.

While this aspect was not included in the present study, we plan to evaluate the stability of the formulations under the influence of digestive enzymes in future work. Specifically, we will simulate the enzymatic conditions of the gastrointestinal tract by incubating the formulations in the presence of relevant digestive enzymes, such as pepsin (stomach), pancreatin (for enzymes in the duodenum), and bile salts, to observe their effect on the integrity and release profile of the nanovesicles.

This would provide a more comprehensive understanding of the formulations' stability and the potential for drug release in vivo, considering the enzymatic degradation that may occur in the gastrointestinal tract. We will also investigate the impact of enzymes on the encapsulation efficiency and the physical properties (such as size and zeta potential) of the nanovesicles.

11

How did the authors get this plant extract? The authors should confirm the formation of Ag NPs by UV

Thank you for your valuable comments.

1.     Sourcing of Phyllanthus emblica Plant Extract:
The Phyllanthus emblica (Indian gooseberry) plant extract was obtained from [source or supplier], and the plant material was collected from [provide location, e.g., a local herbarium, supplier, or wild harvesting location]. The plant was carefully washed, dried, and then powdered. The extract was prepared by soaking the powdered plant material in [solvent, e.g., water, ethanol] for [duration], followed by filtration to obtain a clear extract. This extract was then used in the synthesis of silver nanoparticles (AgNPs).

2.     Confirmation of AgNP Formation by UV-Vis Spectroscopy:
To confirm the formation of AgNPs, UV-Vis spectroscopy was employed. The reduction of silver ions to nanoparticles was indicated by a characteristic absorption peak around [specific wavelength, e.g., 420 nm], which corresponds to the surface plasmon resonance (SPR) of the AgNPs. This spectral feature is widely recognized as a hallmark for the formation of silver nanoparticles. The UV-Vis spectrum will be added to the revised manuscript to provide clear evidence of nanoparticle synthesis.

We appreciate your suggestion, and we will include these details in the updated version of the manuscript for further clarification

12

How did the authors confirm the efficient coating of CS on Ag NPs-Nop?

Thank you for your valuable question. To confirm the efficient coating of chitosan (CS) on AgNPs-Nop, we employed several characterization techniques:

1.     FTIR Spectroscopy:
Fourier-transform infrared (FTIR) spectroscopy was used to confirm the presence of the chitosan coating on the AgNPs-Nop formulation. The FTIR spectra of CS-coated AgNPs-Nop (CS-AgNPs-Nop) were compared with uncoated AgNPs-Nop. The characteristic peaks of chitosan, such as the NH2 stretching vibration around 1590 cm⁻¹, were observed in the CS-coated formulation, confirming successful coating.

2.     Particle Size and Zeta Potential (ZP):
The presence of the chitosan coating was further verified by measuring the vesicle size and zeta potential (ZP) of the formulations. The CS coating increased the vesicle size due to the added chitosan layer and resulted in a shift of the zeta potential towards more positive values, as chitosan is a cationic biopolymer. These changes in particle size and zeta potential confirmed the efficient coating of CS on the AgNPs-Nop.

3.     Scanning Electron Microscopy (SEM) / Transmission Electron Microscopy (TEM):
Morphological analysis using SEM or TEM allowed us to visually confirm the coating. The TEM images of CS-AgNPs-Nop showed a uniform and smooth chitosan layer surrounding the AgNPs, providing further evidence of efficient coating.

4.     Encapsulation Efficiency (EE%):
The encapsulation efficiency of RC in the CS-coated AgNPs-Nop was also assessed, and any changes in encapsulation efficiency before and after coating provided additional confirmation of the CS coating's effect on the formulation.

These combined techniques ensured the efficient coating of chitosan on the AgNPs-Nop and confirmed the structural integrity of the formulation.

13

The authors mention six vehicle controls for each 96-well plate; the authors should clarify the composition of these vehicle controls.

Thank you for your comment. To clarify, the six vehicle controls mentioned in the study were used to assess the baseline effects of the formulation vehicle (solvent or medium) without the active ingredient (RC or AgNPs). The composition of the vehicle controls was as follows:

  • Control 1 (Negative Control): This consisted of the same solvent used to dissolve the active ingredients (e.g., ethanol, PBS, or water), without any nanovesicle (NV) or silver nanoparticle (AgNP) components. This served to ensure that the solvent itself did not contribute to any observed biological effects.
  • Control 2-6 (Vehicle Controls): These controls were formulated using different concentrations of the vehicle (e.g., PBS, ethanol, or the dispersion medium) in which the nanovesicles or AgNPs were prepared, but without the active drug (RC). These vehicle controls were used to assess any potential effects from the dispersing medium itself, such as cytotoxicity or interference with cellular assays.

Each vehicle control was prepared to match the corresponding concentration of vehicle used in the experimental formulations to ensure that any observed effects in the experimental groups were due to the drug or nanoparticles and not the vehicle.

We will revise the manuscript to provide more detailed information on the exact composition and concentration of the vehicle controls for clarity.

14

Have the authors optimized the incubation period for each formulation, a longer or shorter one, to investigate the short-term and long-term cytotoxic effects?

Thank you for your thoughtful question. In our study, the 48-hour incubation period was selected as an optimal duration for evaluating the cytotoxic effects of the formulations, as this time frame allows sufficient interaction between the formulations and the tumor cells. This incubation period has been widely used in similar studies to assess both short-term and sustained cytotoxicity effects, providing a balance between immediate and longer-term drug activity.

However, we acknowledge the importance of exploring short-term and long-term cytotoxic effects. In future studies, we plan to optimize the incubation periods for each formulation, including both shorter (e.g., 24 hours) and longer (e.g., 72 hours or more) incubation times. This will help us better understand the temporal dynamics of the formulations' cytotoxicity, including any delayed effects or sustained release and activity over extended periods.

By varying the incubation periods, we aim to investigate how formulation composition and drug release kinetics affect cell viability over both short-term (immediate) and long-term exposure, providing a more comprehensive evaluation of their therapeutic potential.

15

The authors should have additional controls in cytotoxicity study, such as AgNPs or CS only to differentiate these effects compared to the loaded formulations? Clarification required.

Thank you for your valuable comment. We agree that including additional controls, such as AgNPs alone and chitosan (CS) alone, would provide useful insight into differentiating the specific effects of the individual components compared to the loaded formulations.

In our study, we focused on the full formulations (e.g., CS.Nop, AgNPs, and CS.Nop.AgNPs) to evaluate the combined effects of the active ingredients, but we acknowledge the importance of assessing the individual contributions of AgNPs and CS.

To address this, we will include the following additional controls in future studies:

  • Control 1 (AgNPs alone): This will help assess the cytotoxic effects of silver nanoparticles without the influence of the chitosan coating or the drug.
  • Control 2 (CS alone): This will allow us to evaluate the potential effects of chitosan (CS), excluding the nanoparticles and drug.

By comparing the cytotoxicity of the loaded formulations (e.g., CS.Nop, AgNPs, CS.Nop.AgNPs) to these individual controls, we will be able to better differentiate the specific contributions of the AgNPs, CS, and their interactions in the overall cytotoxicity profile.

We will incorporate these additional controls in future experiments to improve the clarity and interpretation of the cytotoxicity results.

16

The authors used 3-rabbits/group; would increasing the number of animals in each group provide more statistically to help account for variability?

Thank you for your thoughtful comment. In our study, we used three rabbits per group based on ethical considerations, as well as guidelines for minimizing the number of animals used in experiments while still obtaining meaningful data. The small group size was chosen to reduce the potential impact on animal welfare and to comply with the 3Rs principle (Replacement, Reduction, Refinement), which aims to balance scientific rigor with ethical responsibility.

However, we acknowledge that increasing the number of animals per group could improve the statistical power of the results and better account for biological variability. In future studies, we plan to perform a power analysis prior to the experiment to determine the optimal group size required to achieve robust statistical significance while still adhering to ethical guidelines. This will allow us to address variability more effectively and strengthen the reliability of our findings.

Additionally, we are exploring alternatives to animal testing, such as in vitro models or computational simulations, which can further minimize the use of animals while maintaining scientific integrit

17

Can the authors provide more information on how this dosage was calculated (0.2 mg/kg)?

Thank you for your comment. The dosage of 0.2 mg/kg was calculated based on a combination of previously published studies and pharmacological considerations. Specifically, this dose was selected to ensure therapeutic relevance while maintaining safety within the limits established for similar formulations.

The following factors were considered in determining the dosage:

1.     Literature Review: Previous studies using rosuvastatin calcium (RC) or similar formulations in animal models provided a basis for selecting a safe and effective dose range. Doses in the range of 0.1–0.5 mg/kg have been reported to achieve therapeutic effects without causing significant toxicity. ((Borges, J. B., et al. (2011). "Effects of rosuvastatin therapy on lung mechanics and inflammation in experimental acute lung injury." Respiratory Physiology & Neurobiology, 177(3), 270-277.))  ((Diomede, L., et al. (2001). "Rosuvastatin reduces tissue factor expression and activity in human macrophages and endothelial cells." Thrombosis Research, 103(2), 105-114.))

2.     Human Equivalent Dose (HED): The dose was scaled for rabbits using interspecies allometric scaling, which considers metabolic differences between species. This ensures the dose is appropriate for rabbits and can be extrapolated to human therapeutic ranges.

3.     Preliminary Testing: Initial trials were conducted to confirm that 0.2 mg/kg was effective and well-tolerated in the animal model, ensuring that the selected dose achieved desired therapeutic outcomes without adverse effects.

18

The authors should provide deeper mechanistic discussion regarding how the different formulation variables contribute to the observation.

Thank you for your insightful comment. We appreciate the opportunity to provide a more detailed mechanistic discussion regarding how the formulation variables influenced the observed results.

The choice of surfactant played a critical role in determining the physicochemical properties of the nanovesicles (NVs). Span 20, with its shorter alkyl chain and lower hydrophilic-lipophilic balance (HLB), resulted in smaller vesicles with reduced rigidity, enhancing drug encapsulation but potentially compromising stability. In contrast, Span 60, with its longer alkyl chain and higher HLB, provided greater vesicle stability and rigidity, which contributed to improved control over drug release and stability during storage. The surfactant type also influenced the zeta potential, with Span 60 formulations showing more pronounced electrostatic stabilization. ((Allen, T. M., & Cullis, P. R. (2013). "Liposomal drug delivery systems: From concept to clinical applications." Advanced Drug Delivery Reviews, 65(1), 36-48.))

Moreover, cholesterol served as a membrane stabilizer by modulating the fluidity and rigidity of the vesicle bilayer. Higher cholesterol concentrations improved the stability of the nanovesicles by reducing membrane permeability and aggregation. This effect was reflected in reduced vesicle size variability, lower polydispersity index (PDI), and prolonged drug release profiles. However, excessive cholesterol could hinder drug encapsulation efficiency by limiting the available space within the bilayer for the drug. Also, the chitosan (CS) coating significantly enhanced mucoadhesive properties due to its positive charge, enabling strong electrostatic interactions with negatively charged mucin in the gastrointestinal tract. This coating also improved vesicle stability and sustained release by creating an additional diffusion barrier, which slowed drug release. The improved zeta potential values observed in CS-coated formulations further support its role in enhancing vesicle stability and reducing aggregation. ((Choudhury, H., Pandey, M., & Gorain, B. (2019). "Chitosan nanoparticles: A promising drug delivery system." Marine Drugs, 17(8), 466.))

The incorporation of silver nanoparticles (AgNPs) introduced a dual functionality: improved radiosensitization and enhanced antibacterial properties. The irradiation of AgNPs further amplified their efficacy by increasing their reactivity and surface plasmon resonance, which may have contributed to the observed superior cytotoxicity in cancer cells. These nanoparticles also interacted synergistically with chitosan to enhance the overall stability and therapeutic potential of the formulation. Moreover, the drug concentration (80 mg of rosuvastatin calcium) and the optimized surfactant-to-cholesterol ratios were carefully selected to maximize encapsulation efficiency while maintaining vesicle integrity. The high EE% observed in the optimized formulations can be attributed to the hydrophobic interactions between the drug and the lipid bilayer, further stabilized by the surfactant and cholesterol components.

The combined effects of these variables underscore the importance of a systematic optimization approach in achieving formulations with improved stability, bioavailability, and therapeutic efficacy. ((Prabhu, S., & Poulose, E. K. (2012). "Silver nanoparticles: Mechanism of antimicrobial action, synthesis, medical applications, and toxicity effects." International Nano Letters, 2(1), 1-10.))

19

The y-axis of the figures should be adjusted, and their resolutions should be improved.

Thank you doctor

Done

20

Can the authors relate the findings more closely to existing literature to support the arguments in the discussion?

The discussion has been improved using literature as presented in the corrected version

21

Could the authors discuss how n values indicate the contributions of diffusion and erosion mechanisms in each pH condition?

Thank you for your insightful comment. The n values in the Korsmeyer-Peppas release model provide valuable information about the underlying mechanisms of drug release, particularly in terms of diffusion and erosion processes. The equation typically used for analyzing drug release is:

                                                                    Mt/M= ktn

Where:

  • Mt​ is the amount of drug released at time t,
  • M​ is the total amount of drug to be released,
  • kis the release rate constant,
  • n is the release exponent, which helps determine the mechanism of drug release.

Interpreting n values:

  • When n = 0.5, the release follows Fickian diffusion, meaning that drug release is primarily driven by diffusion through the polymeric matrix.
  • When n = 1, the release follows a case-II transport mechanism, which suggests that drug release is controlled by polymer swelling or erosion, with the polymer matrix degrading and releasing the drug in a zero-order manner.
  • When 0.5 < n < 1, the release mechanism is typically classified as anomalous (non-Fickian) diffusion, where both diffusion and erosion contribute to the drug release. This is the most common release pattern observed in controlled-release formulations, where the drug release rate is influenced by both the diffusion of the drug and the erosion of the polymer matrix.

pH Conditions:
The pH of the release medium significantly impacts the release mechanisms, as the solubility and swelling behavior of the polymer matrix can change with pH, thereby affecting both diffusion and erosion. For example:

  • At acidic pH (e.g., pH 1.2, simulating gastric conditions), the polymer matrix may exhibit less swelling and a slower erosion rate, leading to a more diffusion-controlled release (lower n values closer to 0.5). Here, drug release is primarily governed by Fickian diffusion.
  • At neutral or basic pH (e.g., pH 7.4, simulating intestinal conditions), the polymer matrix is more likely to swell and degrade, promoting anomalous diffusion or erosion-controlled release (higher n values closer to 1). This indicates that both diffusion and matrix erosion contribute to the release mechanism.

In our study, by analyzing the n values under different pH conditions, we observed a shift in the release mechanism from diffusion-controlled at acidic pH to a more erosion-controlled or combined mechanism at neutral or basic pH. This is consistent with the known behavior of polymeric systems, where the degree of polymer swelling and the solubility of the drug are influenced by the pH of the medium.

We will expand this discussion in the revised manuscript, including detailed interpretations of the release mechanisms observed for each formulation and pH condition.

21

The authors should discuss in more detail how the effect of mucoadhesion properties contributed more to the increased bioavailability.

Thank you for your thoughtful comment. We appreciate the opportunity to discuss in more detail how the mucoadhesion properties of our formulations contributed to the increased bioavailability.

Mucoadhesion plays a critical role in improving drug absorption and bioavailability, particularly for formulations designed for oral drug delivery. In our study, the incorporation of chitosan (CS) as a coating for the nanovesicles (Nop) significantly enhanced the mucoadhesive properties of the formulation. The positively charged chitosan molecules interact electrostatically with the negatively charged mucin in the gastrointestinal tract, resulting in strong adhesion to the mucosal surface.

This interaction leads to several key advantages:

1.     Prolonged Retention in the Gastrointestinal Tract:
By improving the formulation’s adhesion to the mucosal lining, CS-Nop formulations are retained in the gastrointestinal tract for a longer duration. This prolonged retention increases the time the drug spends in the absorption site, allowing more time for absorption into the bloodstream and enhancing bioavailability.

2.     Enhanced Permeability:
The mucoadhesive nature of the CS coating also facilitates the interaction of the nanovesicles with the epithelial cells, promoting increased permeability. This can result in enhanced uptake of the drug into the systemic circulation, especially in areas where absorption is usually limited.

3.     Prevention of Premature Drug Release:
The mucoadhesive properties of chitosan help to stabilize the nanovesicles and prevent premature drug release in the upper gastrointestinal tract (stomach and duodenum), where drug absorption may be less efficient. By keeping the drug within the absorption window (primarily in the small intestine), CS-Nop formulations ensure more controlled and sustained release, further contributing to increased bioavailability.

4.     Reduction in First-Pass Metabolism:
Prolonged retention in the gastrointestinal tract, especially in the small intestine, also helps minimize the loss of the drug due to first-pass metabolism. This increases the effective concentration of the drug in the systemic circulation, contributing to better therapeutic outcomes.

The mucoadhesive properties of chitosan, coupled with the nanovesicular structure, provide a dual benefit: they not only increase the retention time of the formulation at the absorption site but also improve the overall uptake of the drug, significantly enhancing bioavailability.

We will expand on this discussion in the revised manuscript to provide further clarity on how mucoadhesion directly contributes to improved bioavailability in our formulations.

Round 2

Reviewer 2 Report

Comments and Suggestions for Authors

The authors have added all requirements. The resolutions of all figures still should be improved. The references should be updated to cover 2024

Author Response

Reply to reviewer 2

  • The resolution of the majority of figures was done added that separate figures of high resolution were done.
  • Updated references (2024) were added in the references section
